# Deep representation learning for clustering longitudinal survival data from electronic health records

Jiajun Qiu [1], Yao Hu[1], Li Li[1], Abdullah Mesut Erzurumluoglu[1], Ingrid Braenne[1], Charles Whitehurst [2], Jochen Schmitz[2], Jatin Arora[1], Boris Alexander Bartholdy[1], Shrey Gandhi [1], Pierre Khoueiry[1], Stefanie Mueller[1], Boris Noyvert[1], Zhihao Ding[1], Jan Nygaard Jensen[1] & Johann de Jong [1] ✉

Precision medicine requires accurate identification of clinically relevant patient subgroups. Electronic health records provide major opportunities for leveraging machine learning approaches to uncover novel patient subgroups. However, many existing approaches fail to adequately capture complex interactions between diagnosis trajectories and disease-relevant risk events, leading to subgroups that can still display great heterogeneity in event risk and underlying molecular mechanisms. To address this challenge, we implemented VaDeSC-EHR, a transformer-based variational autoencoder for clustering longitudinal survival data as extracted from electronic health records. We show that VaDeSC-EHR outperforms baseline methods on both synthetic and real-world benchmark datasets with known ground-truth cluster labels. In an application to Crohn's disease, VaDeSC-EHR successfully identifies four distinct subgroups with divergent diagnosis trajectories and risk profiles, revealing clinically and genetically relevant factors in Crohn's disease. Our results show that VaDeSC-EHR can be a powerful tool for discovering novel patient subgroups in the development of precision medicine approaches.

There has been a notable shift in the healthcare sector toward digitizing patient information, with electronic health records (EHRs) emerging as the new norm. As of 2018, the adoption rate of EHR systems has surpassed 84% and 94% in the US and UK, respectively[1,2]. EHR systems offer a comprehensive and easily accessible source of patient data, typically gathering data from millions of individuals over many years, encompassing various sources (such as primary and secondary care) and modalities (such as diagnoses, medications, and lab tests). The extensive nature of EHRs makes them a valuable resource for healthcare research, enabling more accurate modeling of patients and their disease risk, onset, and progression. However, the sheer size and complexity of EHR data inevitably pose challenges to modeling efforts, necessitating the development of sophisticated algorithms and data processing methods. Rapid advancements in the field of deep learning

(DL) have had a profound impact on a wide range of industries and provide many opportunities for improving healthcare as well[3,4]. Specifically, DL has shown great promise in learning meaningful patient representations from EHRs, by virtue of its ability to uncover hidden patterns and trends in such complex datasets. These representations can then be used for a wide variety of downstream tasks such as patient stratification, disease risk prediction, disease progression modeling, etc. For example, CLOUT used long short-term memory (LSTM) networks for learning patient representations from EHRs, which were then used for downstream tasks such as mortality prediction[5]. Leveraging recent advances in large language modeling technology[6,7], Med-BERT and BEHRT were two early examples of using transformer neural networks for learning representations from EHR sequences[3,4], with applications including heart failure prediction and pancreatic cancer

[1]Global Computational Biology and Digital Sciences, Boehringer Ingelheim Pharma GmbH & Co. KG, Biberach an der Riß, Germany. [2]Immunology & Respiratory Diseases, Boehringer-Ingelheim, Ridgefield, CT, USA. ✉e-mail: johanndejong@hotmail.com

prediction. As a more recent example, Placido et al. presented a transformer model for detecting patients with a high risk of pancreatic cancer from EHR data[8]. As a final example, Chung et al. applied the large language model GPT-4 Turbo[9] on procedure descriptions and clinical notes retrieved from EHRs for tasks including the prediction of hospital mortality, hospitalization duration, and ICU duration[10].

Here, we explore the problem of learning clustered patient representations from longitudinal EHR data, in the context of disease-associated risk events. Patient clustering is an important concept in the field of precision medicine. Precision medicine aims to provide the right treatment, to the right patient at the right time, by utilizing individual patient characteristics to guide clinical decision-making, instead of population-wide averages of patient characteristics[11]. Patient clustering supports the development of precision medicine approaches by detecting patterns and trends within a certain patient population of interest, which can serve as the basis for the identification of novel disease subtypes. By studying the causal molecular mechanisms of these disease subtypes, more targeted and personalized therapeutic approaches can be developed. Disease subtyping is often relevant in the context of certain risk events[12]. For example, why do some patients with Crohn's disease, a subtype of inflammatory bowel disease (IBD), progress to intestinal stricture (a narrowing of the intestines due to the formation of scar tissue and muscular hypertrophy), and others do not? As Crohn's disease is a multifactorial disease, it is likely that there are multiple mechanisms associated with progression toward intestinal obstruction[13]. Typically, one of the following two approaches is applied for elucidating how patient subgroups correlate with event risk: (1) Start with identifying patient clusters, and then analyze the risk event within each of the clusters[14]. This approach has inherent limitations because resulting clusters are not guaranteed to correlate to the event risk. (2) Start with stratifying patients by the risk event, and then identify subgroups within the risk strata[15]. The main limitation here is that it can be difficult to identify patient subgroups with differentiating generative mechanisms. In other words, a high-risk patient subgroup could exhibit significant heterogeneity in causal molecular mechanisms, and patient subgroups with comparable survival outcomes could have varying responses to identical treatments[16]. Hence, to more accurately identify novel patient subgroups characterized by both divergent diagnosis trajectories and time-to-event profiles, patient clustering should be integrated with risk modeling (time-to-event analysis) to enable studying their interactions.

While several studies have previously explored patient clustering and risk modeling[17–20], none of these approaches directly integrated risk modeling with clustering. In all cases, any resulting clusters were purely driven by the risk event and would suffer from the limitations we outlined above. These limitations were recently addressed by the introduction of VaDeSC (Variational deep survival clustering)[21], which integrates risk modeling with clustering using a variational autoencoder (VAE) framework. The VAE is a probabilistic generative model where observations are assumed to originate from latent representations sampled from a prior distribution of choice. This prior distribution can function as a regularizer on the learned representations by enforcing a certain latent structure[22]. In the original formulation of the VAE, the researchers explored multivariate Gaussian and multivariate Bernoulli priors[22]. More recently, with models such as Variational Deep Embedding (VaDE)[23], Gaussian mixture priors have been studied for the purpose of enforcing a latent structure that promotes clustering. VaDeSC combines a Gaussian mixture prior as used in VaDE with a Weibull mixture distribution for modeling cluster-specific survival, to learn cluster-specific associations between covariates and survival times.

Given the importance of risk modeling and clustering in healthcare research, methods such as VaDeSC provide many opportunities for applications within the healthcare domain. Specifically, insights into patients' disease history and progression are of great importance for developing a more comprehensive disease understanding and eventually developing more effective and personalized therapies[24,25]. Hence, in this work, we developed a novel application of VaDeSC by building upon recent advances in patient representation learning from EHR sequences[4]. We integrated VaDeSC with a custom-designed autoencoding transformer-based architecture and implemented VaDeSC-EHR, a first attempt at disentangling the complex relationships between cluster-specific longitudinal disease histories and event risk profiles as retrieved from EHRs. In this study, we demonstrate VaDeSC-EHR's ability to capture statistical interactions between cluster-specific diagnosis trajectories and survival times, enabling it to discover novel and clinically relevant patient subgroups.

## Results
### VaDeSC-EHR architecture
VaDeSC-EHR takes a patient's disease history (diagnosis sequence) as an input and maps it into a latent representation $z$ using a transformer-based VAE with a Gaussian mixture prior (Fig. 1). Adopting the VaDeSC approach[21], the survival outcome is modeled using a mixture of Weibull distributions with cluster-specific parameters $\beta$. The parameters of the Gaussian mixture and Weibull distributions are jointly optimized using the diagnosis sequences and survival outcomes. Note that in this work, we use the terms survival modeling, risk modeling, and time-to-event modeling interchangeably, as we do with patient cluster, patient subgroup, and patient subpopulation. More details about the model architecture and loss function can be found in the "Methods" section.

### UK Biobank datasets
In this study, we applied VaDeSC-EHR to EHR data from UK Biobank (Table 1, Supplementary Table 1, and Supplementary Fig. 1). We mapped the Read v2/3 diagnosis codes and ICD-9 codes (International Classification of Diseases, 9th revision) provided by UK Biobank to ICD-10 codes (International Classification of Diseases, 10th revision; details in the "Methods" section) and designed a multi-level ICD-10 diagnosis embedding layer to capture the hierarchical nature of the ICD-10 ontology. In this embedding layer, each diagnosis is represented by a combination of six distinct embeddings (Fig. 2): three for the ICD-10 code (subcategory, category, and block), one for age, one for type, and one for position. The age embedding represents the patient's age at the time of diagnosis and can additionally assist the model in understanding the temporal gaps between diagnoses. The type embedding differentiates between diagnoses derived from primary care data and those from hospital data. The position embedding, representing visits, establishes the relative placement of diagnoses within the diagnosis sequence, allowing the network to recognize positional relationships between diagnoses. Diagnoses originating from the same visit will have identical position embeddings.

### VaDeSC-EHR outperforms baseline methods on synthetic data
In order to establish confidence in the methodology, we first technically validated VaDeSC-EHR on a synthetic benchmark dataset generated using a VaDeSC-EHR decoder (see "Methods" section) and compared its generalization performance to a range of baseline methods (Supplementary Figs. 2 and 3). The baseline methods were: variational deep survival clustering with a multilayer perceptron (VaDeSC-MLP), semi-supervised clustering (SSC)[17], survival cluster analysis (SCA)[18], deep survival machines (DSM)[19], and recurrent neural network-based DSM (RDSM)[20], as well as k-means and regularized Cox PH as naïve baselines.

Using synthetic data provides optimal control over the data-generating process and allows the generation of ground-truth cluster labels, which are not used for training the model, but can be used post hoc to unambiguously assess generalization performance. It is important to note that no baseline methods explicitly designed for clustering longitudinal survival data are currently available from the

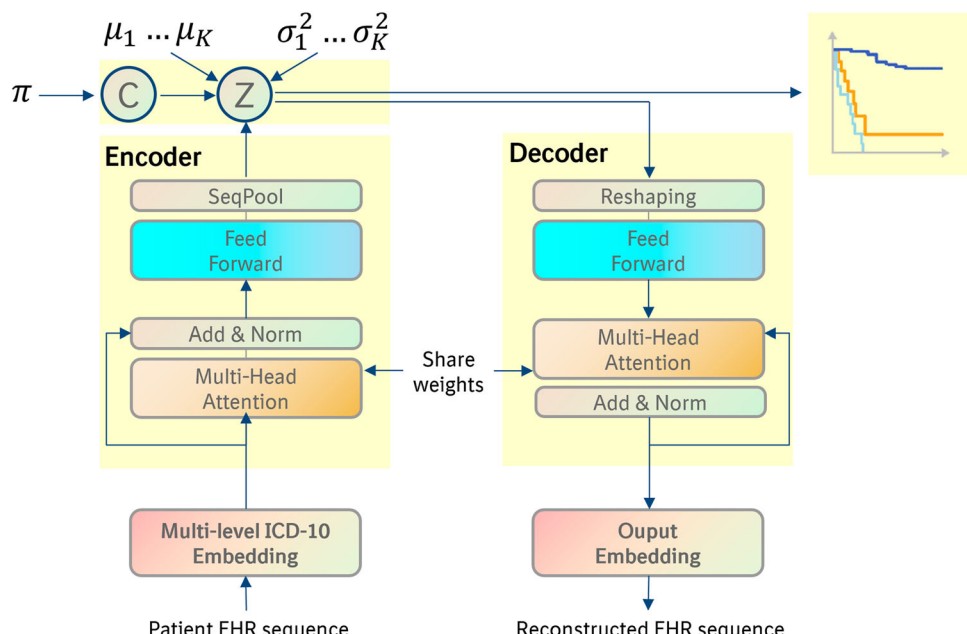

**Fig. 1 | Architecture of VaDeSC-EHR.** First, an embedding is computed for a patient's diagnosis sequence. This embedding serves as input for multiple transformer blocks, where the final pooling layer generates the latent representation Z for the patient. Z is regularized toward a Gaussian mixture distribution by including a variational term in the loss function. Z is then used to predict the time-to-event[21], as well as passed to the transformer decoder for reconstructing the patient's diagnosis sequence. Weights are shared between the encoder and decoder. For more details, please refer to the "Methods" section.

**Table 1 | Summary of the datasets used in this study**

|  |  | UK Biobank EHR | T1D/T2D benchmark | CD application |
|---|---|---|---|---|
| Source | Primary care | 5,999,672 | 60,148 | 53,470 |
|  | Hospital care | 3,434,668 | 68,628 | 39,179 |
| Age | 0–30 | 219,721 | 937 | 1920 |
|  | 30–60 | 5,468,079 | 52,645 | 51,673 |
|  | 60– | 3,746,540 | 75,194 | 39,056 |
| Sex | Male | 157,065 | 1459 | 825 |
|  | Female | 195,826 | 865 | 1083 |
| Right censored? | Yes |  | 999 | 1591 |
|  | No |  | 1325 | 317 |
| Diagnosis sequence length (years) |  | 18.75(20.83) | 16.04(12.73) | 19.17(21.10) |
| Interval between visits (years) |  | 0.46(0.96) | 0.34(0.66) | 0.40(0.81) |
| Number of records |  | 9,434,340 | 128,776 | 92,649 |
| Number of patients |  | 352,891 | 2324(T1D: 494; T2D: 1830) | 1908 |

For diagnosis sequence length (years) and interval between visits (years), we show the median(IQR).
*CD* Crohn's disease, *T1D* type 1 diabetes, *T2D* type 2 diabetes.

| Position embedding | 0 | 0 | 0 | 1 | 2 | 3 | 3 | 4 |
|---|---|---|---|---|---|---|---|---|
| Type: Primary/Hospital | Primary | Primary | Primary | Hospital | Hospital | Hospital | Hospital | Hospital |
| Age | 36.33 | 36.33 | 36.33 | 42.25 | 48.00 | 53.58 | 53.58 | 60.22 |
| ICD10: Block | R50-R69 | K50-K52 | O30-O48 | E65-E68 | Z80-Z99 | I10-I15 | I32-I52 | E50-E64 |
| ICD10: category | R63 | K50 | O43 | E66 | Z90 | I10 | I51 | E53 |
| ICD10: subcategory | R635 | K509 | O431 | E660 | Z904 | I110 | I516 | E538 |
|  | ● | ● | ● | ● | ● | ● | ● | ● |
|  | Diag. 1 | Diag.2 | Diag.3 | Diag.4 | Diag.5 | Diag.6 | Diag.7 | Diag.8 |

**Fig. 2 | The embedding structure.** Simulated example of a diagnosis sequence for a single individual with 8 diagnoses spread across 5 primary or hospital care visits (0–4) embedded at three levels of the ICD-10 ontology. Diag.: Diagnosis.

literature, and certain adaptations in applying these baseline methods need to be made (see "Methods" section). In addition to evaluating the cluster predictions using the balanced accuracy (ACC), normalized mutual information (NMI) and adjusted Rand index (ARI), we evaluated the time-to-event predictions using the concordance index (CI). Note that due to the noise in the data-generating process, achieving perfect performance was not possible.

As can be seen in Table 2, VaDeSC-EHR significantly outperformed all other methods in retrieving the ground-truth clusters from the benchmark data. Even when replacing the transformer encoder/decoder architecture with a basic multilayer perceptron (MLP) encoder/decoder with simple Term Frequency-Inverse Document Frequency (TF-IDF) features (VaDeSC-MLP), performance is still significantly better, highlighting the expressiveness provided by jointly modeling clustering and event risk using the VaDeSC approach. As expected, the transformer architecture helped VaDeSC-EHR to achieve much better performance on the longitudinal clustering task than VaDeSC-MLP (VaDeSC-EHR: ACC = 0.64 ± 0.04, NMI = 0.72 ± 0.04 and ARI = 0.66 ± 0.07; VaDeSC: ACC = 0.57 ± 0.05, NMI = 0.37 ± 0.04, and ARI = 0.44 ± 0.03, Table 1). Importantly, VaDeSC-EHR achieved its superior clustering performance while hardly sacrificing performance on the risk prediction task. VaDeSC-MLP only showed marginally better performance on survival prediction than VaDeSC-EHR (CI = 0.79 ± 0.02 for VaDeSC-MLP vs. CI = 0.77 ± 0.01 for VaDeSC-EHR, p-value = 0.007). Given the known regularizing effects of variational loss terms[22], it is likely that VaDeSC-EHR's risk prediction performance could be improved (possibly at the expense of some clustering performance) by weighting the different loss terms (e.g. beta-VAE[26]). We leave this to future study.

The remaining models (k-means+Cox PH, SSC, SCA, DSM, and RDSM) performed no better than random at retrieving the ground-truth clusters. Whereas RDSM clearly stood out as the third-best performing on the risk prediction task, it did perform worse than VaDeSC-MLP. This is interesting, because RDSM explicitly models EHR data as sequences, whereas VaDeSC-MLP does not. However, as explained in the introduction, the RDSM clusters are solely driven by survival times. This can explain the observed difference between VaDeSC-MLP and RDSM on the risk prediction task, and again highlights the importance of modeling the interactions between the diagnosis sequences and the survival times.

## VaDeSC-EHR outperforms baseline methods on a T1D/T2D benchmark

While VaDeSC-EHR outperformed all baseline methods on synthetic data, we still wondered how VaDeSC-EHR would compare to the baseline methods on real-world data instead of data generated through simulation. For this reason, we designed a second experiment for technically validating VaDeSC-EHR, using real-world data from UK Biobank instead of synthetic data: distinguishing 494 type 1 diabetes (T1D) patients from 1830 type 2 diabetes (T2D) patients in their progression toward retinal disorders. We chose diabetes for three main reasons. First, like for the synthetic data, it allowed us to unambiguously define ground-truth cluster labels (T1D and T2D), which would not be used for training the model but could be used post hoc to assess generalization performance. Second, with a censoring rate of 42% (Table 1), the dataset is relatively balanced allowing for a sufficiently powered comparison between the methods. Third, besides some similarities[27,28], there are clear and known differences in clinical course and pathophysiological mechanisms between T1D and T2D[29], which can be exploited by machine learning algorithms in their attempts to recover the ground-truth cluster labels.

We compared VaDeSC-EHR's generalization performance to a range of baseline methods using nested cross-validation (NCV)[30], by performing 10 runs using the same hyperparameters but with differently randomly initialized embeddings and neural network parameters, confirming the stability of the results (Supplementary Fig. 4). Note that the T1D and T2D labels were not used for clustering, and thus also the age at first diabetes diagnosis was not available to the model. To ensure that no information leakage occurred due to the inclusion of age-related complications, we additionally performed an analysis taking age information relative to the age at the start of the diagnosis sequence (VaDeSC-EHR_relage), by subtracting the age at first diagnosis from all elements in the age sequence. The performance of VaDeSC-EHR_relage was almost identical to that of VaDeSC-EHR, indicating that VaDeSC-EHR does indeed primarily utilize the time intervals between diagnoses, rather than the age itself.

The main observations from our method comparison strongly mirror those made from the synthetic benchmark. Again, VaDeSC-EHR outperformed the other methods at retrieving the ground-truth clustering (Fig. 3 and Table 3; AUC: 0.81 ± 0.01, ACC = 0.81 ± 0.02), while not giving up any risk prediction performance. A UMAP projection of the latent representations of the diagnosis sequences indeed showed that patients with the same ground-truth disease label (either T1D or T2D) tended to be close in the latent space (Supplementary Fig. 5). VaDeSC-MLP (ACC: 0.71 ± 0.09) was the next best-performing method, performing better than VaDeSC-EHR_nosurv (ACC: 0.64 ± 0.06), in which the survival loss was turned off. VaDeSC-EHR's performance was degraded when not considering the survival times (VaDeSC-EHR_nosurv), illustrating that in achieving its superior performance, the full VaDeSC-EHR model did indeed exploit the interactions between EHR trajectories and survival times. As opposed to the synthetic benchmark, the performance of VaDeSC-EHR and VaDeSC-MLP on the risk prediction task was statistically indistinguishable (CI: 0.72 ± 0.07, p-value: 0.69). Mirroring the results on the synthetic benchmark, all other models performed poorly at retrieving the ground-truth clusters. Finally, RDSM again clearly stood out as the third-best method on

## Table 2 | Performance on synthetic benchmark data

| Method | ACC | NMI | ARI | CI |
|---|---|---|---|---|
| *k*-means +Cox PH | 0.334 ± 0.002*(0.008) | 0.0003 ± 0.0001*(0.008) | -0.0003 ± 0.0003*(0.008) | 0.505 ± 0.001*(0.008) |
| SSC | 0.3334 ± 0.0004*(0.008) | 0.0003 ± 0.0002*(0.008) | 0.00005 ± 0.00050*(0.008) | |
| SCA | 0.336 ± 0.007*(0.012) | 0.03 ± 0.02*(0.012) | 0.006 ± 0.040*(0.012) | 0.639 ± 0.006*(0.008) |
| DSM | 0.3327 ± 0.0004*(0.008) | 0.0005 ± 0.0005*(0.008) | -0.0005 ± 0.001*(0.008) | 0.49 ± 0.01*(0.008) |
| RDSM | 0.502 ± 0.002*(0.007) | 0*(0.007) | 0*(0.007) | 0.500 ± 0.001*(0.008) |
| VaDeSC-MLP | 0.57 ± 0.05*(0.008) | 0.37 ± 0.04*(0.008) | 0.44 ± 0.03*(0.008) | 0.79 ± 0.02*(0.008) |
| VaDeSC-EHR | 0.64 ± 0.04 | 0.72 ± 0.04 | 0.66 ± 0.07 | 0.77 ± 0.01 |

Comparison between VaDeSC-EHR and other methods adapted for clustering longitudinal survival data, in terms of balanced accuracy (ACC), normalized mutual information (NMI), adjusted Rand index (ARI), and concordance index (CI). Reported ± is one standard deviation and the significance of the difference (p-value < 0.05) between VaDeSC-EHR and the other methods is indicated by an asterisk. Significance is based on a two-sided Mann–Whitney U test. Detailed p-values are shown in the brackets.

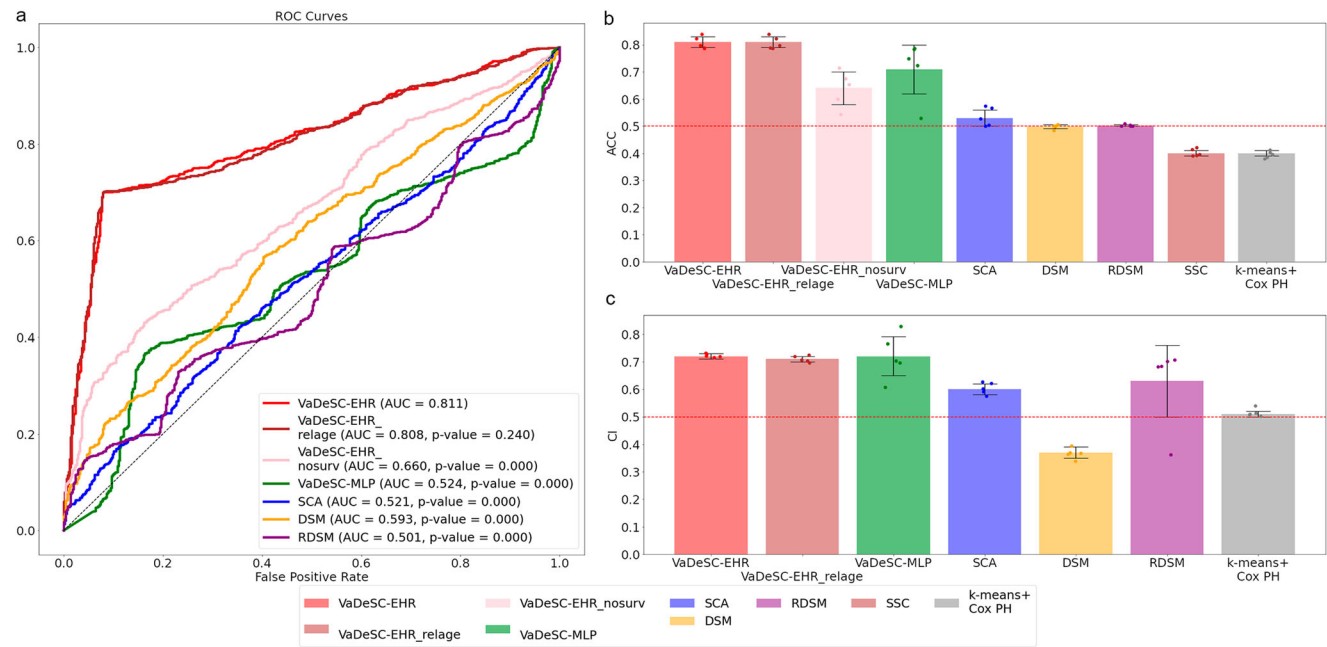

**Fig. 3 | Generalization performance on synthetic benchmark data.** Comparison between VaDeSC-EHR and the other methods used for clustering longitudinal survival data. VaDeSC-EHR_nosurv represents VaDeSC-EHR trained without risk Loss. VaDeSC-EHR_relage represents VaDeSC-EHR trained with age information taken relative to the age at the start of the diagnosis sequence. The analyses are based on nested 5-fold cross-validations ($n = 5$). **a** Performance on retrieving the ground-truth clustering, in terms of the area under the receiver-operating characteristic (ROC), with $p$-values for the significance of the difference between VaDeSC-EHR and the other methods. Significance is assessed using a permutation test combined with 10,000 bootstrap iterations. **b** Performance on retrieving the ground-truth clustering in terms of balanced accuracy (ACC), with 0.5 for random performance. **c** Performance on time-to-event prediction, in terms of concordance index (CI), with 0.5 for random performance. Data are presented as mean values ± standard deviation.

### Table 3 | Generalization performance on T1D/T2D benchmark data

| Method | ACC | NMI | ARI | CI |
|---|---|---|---|---|
| *k*-means+Cox PH | 0.40 ± 0.01*(0.008) | 0.06 ± 0.01*(0.008) | 0.09 ± 0.01*(0.008) | 0.51 ± 0.01*(0.008) |
| SSC | 0.40 ± 0.01*(0.008) | 0.06 ± 0.02*(0.008) | 0.087 ± 0.009*(0.008) | |
| SCA | 0.53 ± 0.03*(0.008) | 0.005 ± 0.006*(0.008) | 0.02 ± 0.02*(0.008) | 0.60 ± 0.02*(0.008) |
| DSM | 0.498 ± 0.007*(0.011) | 0.0003 ± 0.0005*(0.011) | 0.003 ± 0.009*(0.011) | 0.37 ± 0.02*(0.008) |
| RDSM | 0.503 ± 0.003*(0.011) | 0.002 ± 0.002*(0.011) | 0.006 ± 0.008*(0.011) | 0.63 ± 0.13*(0.008) |
| VaDeSC-MLP | 0.71 ± 0.09*(0.016) | 0.15 ± 0.09(0.222) | 0.13 ± 0.14(0.548) | 0.72 ± 0.07(0.690) |
| VaDeSC-EHR_nosurv | 0.64 ± 0.06*(0.008) | 0.06 ± 0.05*(0.008) | 0.08 ± 0.03*(0.008) | |
| VaDeSC-EHR_relage | 0.81 ± 0.02(0.458) | 0.24 ± 0.04(0.458) | 0.23 ± 0.04(0.458) | 0.71 ± 0.01*(0.047) |
| VaDeSC-EHR | 0.81 ± 0.02 | 0.24 ± 0.04 | 0.23 ± 0.04 | 0.72 ± 0.01 |

Comparison between VaDeSC-EHR and the other methods used for clustering longitudinal survival data, in terms of balanced accuracy (ACC), normalized mutual information (NMI), adjusted Rand index (ARI), and concordance index (CI). Reported ± is one standard deviation, and the significance of any difference ($p$-value < 0.05) between VaDeSC-EHR and the other methods is indicated by an asterisk. Significance is based on a two-sided Mann–Whitney $U$-test. Detailed $p$-values are shown in the brackets.

the risk prediction task (CI = 0.63 ± 0.13), highlighting the importance of accounting for interactions between diagnosis sequences and survival times.

Concluding, VaDeSC-EHR outperformed the baseline methods not only on synthetic data but also on an experiment designed from real-world data. This reinforced our confidence in the approach and again highlighted the potential of integrating VaDeSC into a transformer-based patient representation learning approach.

### Application: VaDeSC-EHR identifies clinically and genetically relevant Crohn's disease patient subgroups

Having gained confidence in the method through two technical validation experiments with ground-truth cluster labels, we applied VaDeSC-EHR to 1908 Crohn's disease (CD) patients from UK Biobank. Here, the purpose was to identify potentially novel patient subgroups related to progression toward intestinal obstruction, an important CD complication[31] that is enriched in CD patients relative to the general population (Supplementary Fig. 6). Here, VaDeSC-EHR identified four patient subgroups (CI: 0.91 ± 0.02) that demonstrated both divergent longitudinal disease histories and divergent risk profiles (Fig. 4). The results were stable across 10 randomly initialized runs (Supplementary Fig. 7). As expected, we observed that patients clustered together tended to be close in the latent space, i.e. have diagnosis trajectories that are similar (Fig. 4a). Additionally, the four patient subgroups each demonstrated distinct time-to-event profiles (Fig. 4b), with the fastest progressing cluster 2 being most strongly enriched for intestinal obstruction (Supplementary Fig. 8a). We found the four clusters to be significantly associated with age of onset ($p$-value: $6.82 \times 10^{-05}$), sex ($p$-value: $2.35 \times 10^{-10}$), genetic principal component number 1 (PC1) ($p$-value: 0.01), but not with UKB recruitment location or overall diagnosis

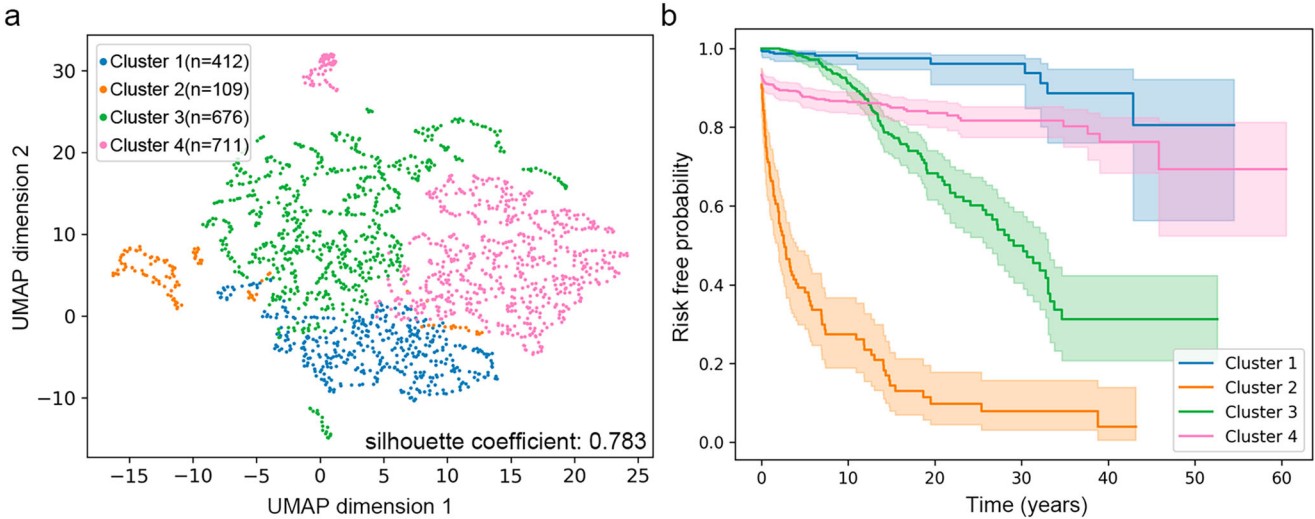

**Fig. 4 | Clustering CD patients from UK Biobank in their progression toward intestinal obstruction. a** UMAP (Uniform Manifold Approximation and Projection) projection of the latent representations of the CD patients, coloring patients by cluster (silhouette coefficient: 0.783). **b** Cluster-specific Kaplan–Meier curves with 95% confidence intervals. Lines denote mean values and shaded regions are 95% confidence intervals.

sequence length (total number of diagnosis codes) (Supplementary Figs. 8–10).

**CD patient subgroups demonstrate distinct disease histories.** To gain insight into the generative mechanisms giving rise to the observed differences in intestinal obstruction risk, we first analyzed differential enrichment of individual diagnoses across the clusters, while adjusting for potential confounding as outlined in the "Methods" section. For example, corroborating previous studies[32–34], we found that many important CD-related ICD-10 codes were reported significantly less frequently in the most slowly progressing cluster 1, such as R10 (abdominal and pelvic pain, adjusted $p$-value: 0.01), R11 (nausea and vomiting, adjusted $p$-value: 0.0007), and D64 (anemia, adjusted $p$-value: 0.0001) (Supplementary Fig. 11). To take full advantage of the longitudinal nature of VaDeSC-EHR we then wanted to analyze how diagnosis trajectories longitudinally differed among patient clusters. Although there was no significant difference in overall diagnosis sequence length between the clusters (Supplementary Fig. 8e), the diagnosis history leading up to the first CD diagnosis was twice as long for the slowly progressing clusters (clusters 1 and 4) as it was for fast progressing clusters (clusters 2 and 3) (Supplementary Fig. 8f), with many diagnosis subsequences significantly enriched in the slow progressors before their first CD diagnosis relative to the fast progressors (Supplementary Fig. 12a). These enriched diagnosis subsequences contained many known comorbidities of CD such as anemia[34], hypertension[35], and hernia[32]. While clusters 1 and 4 were strongly enriched for comorbidities before the first CD diagnosis, there was no significant enrichment of comorbidities after the first CD diagnosis, except for hypertension. On the other hand, clusters 2 and 3 were hardly significantly enriched for comorbidities leading up to their first CD diagnosis (Supplementary Fig. 12b), but these patients did appear to progress more rapidly in their disease, as evident from comorbidities developing after CD onset, including abdominal pain, nausea, vomiting and iron deficiency anemia[32,36,37] (Supplementary Fig. 12b). Interestingly, although patients in clusters 1 and 4 progressed more slowly, their diagnosis trajectories were nonetheless quite distinct (Supplementary Fig. 13). For example, cluster 4 was more enriched for hypertension before first the CD diagnosis, whereas cluster 1 was more enriched for pain[38,39] (back pain, abdominal pain and pain in joint), and respiratory abnormalities[40,41] (cough and asthma) (Supplementary Figs. 13 and 14).

**Does smoking protect against progression toward intestinal obstruction in some CD patients?** Because of the previously reported interesting and poorly understood differential effects of smoking behavior between the two main types of inflammatory bowel disease (protective in ulcerative colitis and harmful in Crohn's disease)[42,43], we were then interested to see whether smoking was uniformly harmful across our identified CD patient clusters. Corroborating previous work[35], we observed that smoking behavior was associated with faster progression toward intestinal obstruction in the overall CD patient population (ever smoked, log hazard ratio: 0.08, Fig. 5a; ICD-10 F17.2: nicotine dependence, log hazard ratio: 0.20, Fig. 5b). Interestingly however, we observed that the slowest progressing cluster 1 was enriched for smoking behavior (ever smoked, $p$-value: $1.86 \times 10^{-05}$; ICD-10 F17.2: nicotine dependence, $p$-value: 0.001) (Fig. 5c, d). Even more surprisingly, within cluster 1, smoking was in fact significantly associated with slower progression toward obstruction, i.e. CD patients who had ever smoked generally progressed more slowly toward intestinal obstruction (ever smoked, log hazard ratio: −0.76, $p$-value: $1.06 \times 10^{-07}$, Fig. 5a; ICD-10 F17.2: nicotine dependence, log hazard ratio: −0.96, $p$-value: 0.001, Fig. 5b). Similarly, "Pack years of smoking" and "Pack years adult smoking as a proportion of life span exposed to smoking" were associated with slower progression toward intestinal obstruction within cluster 1, as were previous and current smoker status (Supplementary Fig. 15). These results could suggest that in CD, as in ulcerative colitis[42,43], patient subgroups exist for which smoking protects against progression toward intestinal obstruction.

**VaDeSC-EHR can identify subgroup-specific molecular mechanisms relevant to drug discovery and precision medicine.** In addition to the EHR data that was used for identifying the four CD patient subgroups presented in this section, UK Biobank provides a wealth of other types of data on the same CD patients that we did not use for clustering. Specifically, the genetics data available for approximately 500,000 patients in the UK Biobank provides interesting opportunities for further characterizing our patient clusters and gaining insight into the generative mechanisms underlying the observed differences between the clusters.

Using the available genetics data, we first computed pathway PRSs (pathway-based polygenic risk scores) to assess differences in the underlying genetic background between fast and slow progression clusters (Supplementary Data 1). Among others, we found that the fast

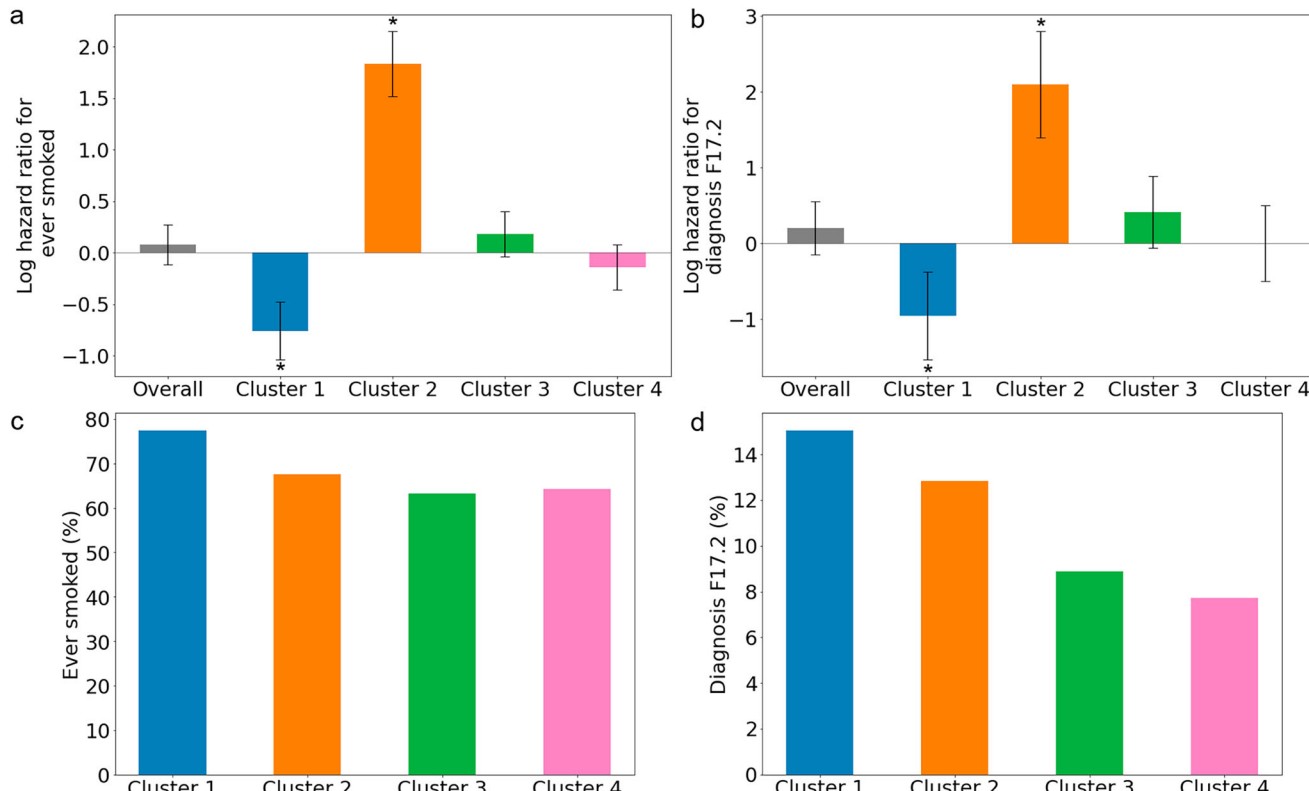

**Fig. 5 | Association of CD clusters with smoking behavior.** The analyses are based on 1,908 CD patients ($n = 1908$). **a** Association of ever having smoked (UK Biobank data field: 20160) with risk of intestinal obstruction (*p*-values are Overall: 0.45, Cluster 1: $1.06 \times 10^{-07}$, Cluster 2: $1.23 \times 10^{-29}$, Cluster 3: 0.109, Cluster 4: 0.200) and **b** Association of nicotine dependence with risk of intestinal obstruction (ICD-10 code: F17.2) (*p*-values are Overall: 0.26, Cluster 1: 0.001, Cluster 2: $4.23 \times 10^{-09}$, Cluster 3: 0.09, Cluster 4: 0.99). Data are presented as mean values ± 95% confidence intervals. They are estimated using two-sided Cox proportional hazards regression models. And the asterisk represents the significance of the *p*-value < 0.05 (multivariate Cox regression). **c** Percentage of patients who ever smoked (*p*-value: $1.86 \times 10^{-05}$) and **d** Percentage of patients with nicotine dependence (ICD-10 code: F17.2) (*p*-value: 0.001). Significance is assessed using multinomial logistic regression with log-likelihood ratio test, which is a two-sided test.

progressors (clusters 2 and 3) displayed a higher genetic burden in a pathway related to the adaptive immune response (Fig. 6a), relative to the slow progressors (clusters 1 and 4) (Fig. 6a). Specifically, we found that the patients in the fastest progressing cluster 3 displayed a higher genetic burden in this pathway than the patients in all other, more slowly progressing, clusters. The involvement of innate immunity versus adaptive immunity in the pathogenesis of CD is an important topic of research[44]. Recently, several studies demonstrated the role of an abnormal adaptive immune response in the pathogenesis of CD[44–46], and the over-reactive adaptive immune response is in fact the target of current CD treatments[46]. Our results confirm the important role that the adaptive immune response may play in the pathogenesis of CD in at least a subpopulation of CD patients.

We then wanted to see if we could identify any specific genetic variants in the adaptive immune response pathway that could potentially explain the significant association of its pathway PRS with our patient subgroups. Among the 150 SNPs, we found SNP rs2523608 to be significantly associated with fast progression toward intestinal obstruction (clusters 2 and especially 3) (Fig. 6b, c). SNP rs2523608 has previously been associated with gastrointestinal disorders, such as celiac disease[47] (known to be bidirectionally causally related to CD[48]), intestinal malabsorption[49] (a common complication of CD[50]), and CD itself[51].

Interestingly, rs2523608 has been reported to be negatively associated with the binding antibody response to interferon beta-1a (IFN beta-1a) therapy in multiple sclerosis (MS), which shares common pathogenic processes with CD[52]. IFN beta-1a is an approved treatment for MS[53] and has been investigated as a potential treatment for CD as well, due to its ability to down-regulate the expression of interleukin-12, a cytokine that is thought to be involved in mucosal degeneration in CD[54]. Unfortunately, there was no significant difference in efficacy between patients receiving IFN beta-1a and those receiving placebo[54]. However, the results of the MS study suggest that one reason for this observed lack of efficacy could lie in the genetic variability across patients. More specifically, depending on the presence of the rs2523608 SNP (risk allele frequency in UKB: 0.42 in CD patients and 0.40 in the general population) in a CD patient, response to IFN beta-1a treatment could be impaired. Using VaDeSC-EHR, we identified two, relatively fast progressing, CD subgroups enriched for this SNP. This demonstrates that VaDeSC-EHR is able to identify subgroup-specific molecular mechanisms relevant to drug discovery and precision medicine, using only longitudinal diagnosis data as input.

## Discussion

Precision medicine aims to develop therapies that are targeted to specific patient subgroups based on their predicted disease risk or progression, or treatment response. Due to their scale, comprehensive nature, and multimodality, EHRs have emerged as a valuable source of data for healthcare research in general and the development of precision medicine approaches specifically[55]. An important concept in precision medicine is the identification of novel patient subgroups, i.e. patient clustering. Retrospective analysis and clustering of EHR trajectories can provide an important first step in the development of precision medicine approaches by supporting the identification of

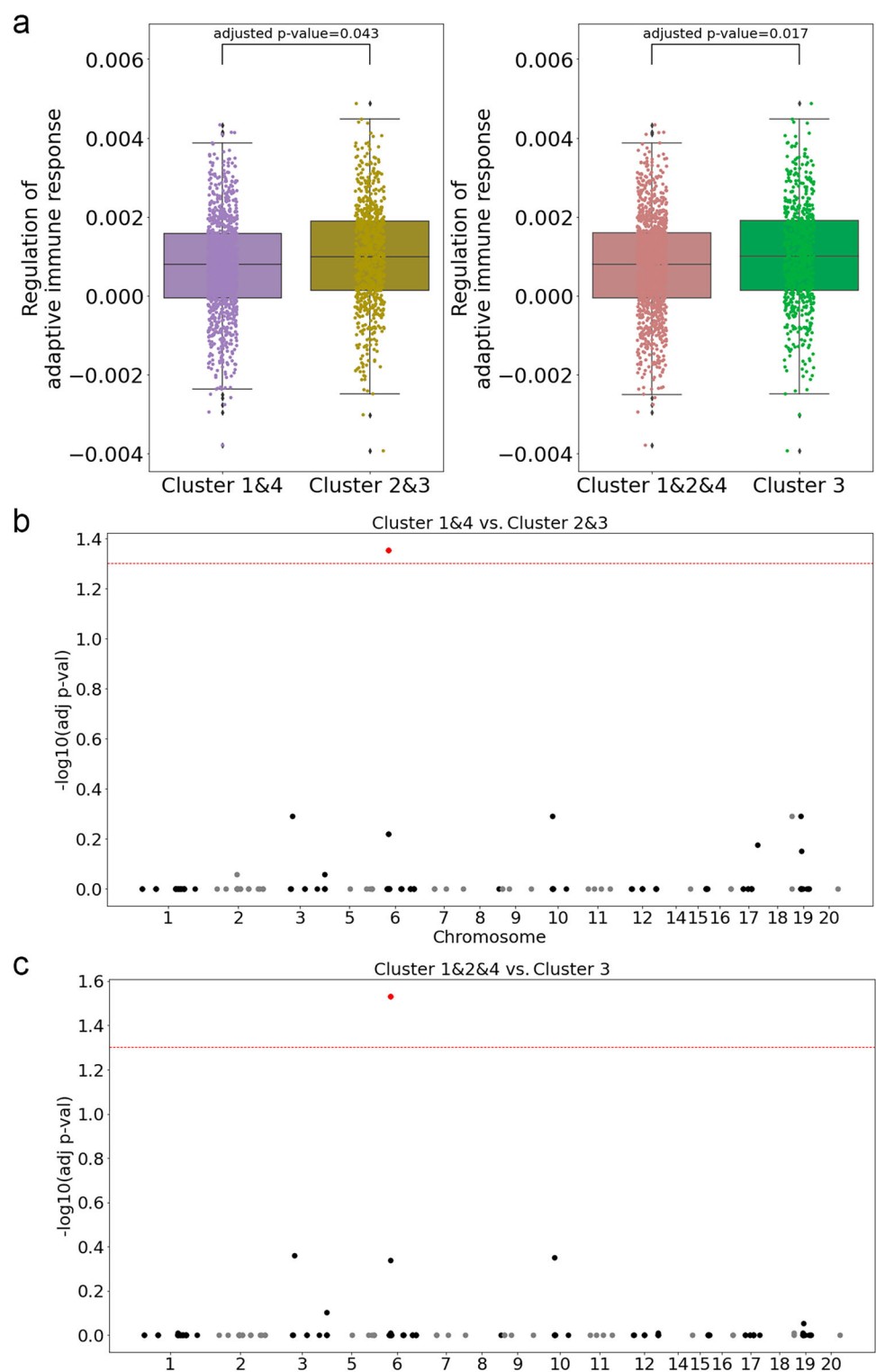

**Fig. 6 | Association of CD clusters with genetics.** The analyses are based on 1908 CD patients (*n* = 1908). **a** Pathway polygenic risk scores (pathway PRS) of the adaptive immune response pathway, comparing clusters 2 and 3 with clusters 1 and 4 (left), and cluster 3 with the other three clusters (right). The bounds of the box are defined by the lower quartile (25th percentile) and the upper quartile (75th percentile). The whiskers extend from the box and represent the data points that fall within 1.5 times the interquartile range (IQR) from the lower and upper quartiles. Any data point outside this range is considered an outlier and plotted separately. Significance is assessed using logistic regression with log-likelihood ratio test, which is a two-sided test. And

multiple testing is corrected using the Benjamini–Hochberg procedure. **b** Enrichment of individual genetic variants used in the pathway PRS in clusters 1 and 4 relative to clusters 2 and 3, with the horizontal line indicating the significance level (FDR adjusted) *p*-value: 0.05. The variant highlighted in red is rs2523608. **c** Enrichment of individual genetic variants used in the pathway PRS in clusters 1, 2, and 4 compared to cluster 3, with the horizontal line indicating the significance level (FDR adjusted) *p*-value: 0.05. The variant highlighted in red is rs2523608. Significance is assessed using a two-sided Chi-square test and multiple testing is corrected using the Benjamini–Hochberg procedure.

novel subgroup-specific and targetable disease mechanisms. Following the identification of a potentially targetable patient subgroup, predictive modeling could be used for prognostic enrichment of clinical trials[56], which can eventually influence treatment decisions in clinical practice, through a drug label reflecting the enrichment strategies used to select patients in the clinical trials[57].

In this paper, we implemented a novel application of VaDeSC[21] leveraging recent advances in the domain of EHR modeling. Specifically, we integrated VaDeSC into a custom-designed transformer-based autoencoding architecture for patient representation learning and developed VaDeSC-EHR, for clustering longitudinal time-to-event data extracted from EHRs. VaDeSC-EHR can exploit statistical interactions between a patient's longitudinal disease history and survival time, enabling it to identify non-trivial patient subgroups characterized by both divergent disease histories and survival times. We validated VaDeSC-EHR on two benchmark experiments with known ground-truth cluster labels, showing that VaDeSC-EHR outperformed all baseline methods on simultaneously the risk prediction task and the task of retrieving the ground-truth clustering. The baseline methods failed to identify clinically relevant subgroups due to either their focus on purely risk-driven clustering or their inability to model EHRs as sequences. As such, our validation highlights the need for novel methods such as VaDeSC-EHR that can more accurately identify novel patient subgroups by modeling the interactions between longitudinal diagnosis trajectories and event risk. After the technical validation, we applied VaDeSC-EHR to the problem of clustering Crohn's disease patients in their progression toward intestinal obstruction. We demonstrated that the resulting subgroups were both clinically and biologically relevant. Specifically, using only diagnosis trajectories as input, VaDeSC-EHR was able to identify molecular mechanisms relevant to only specific subgroups of CD patients. Knowledge of such mechanisms is critical for drug discovery and the development of precision medicine approaches.

VaDeSC-EHR is a first attempt at integrating recent advances in EHR trajectory clustering and risk modeling and we see many opportunities for future research, a few of which we list here. First, the attention mechanism in the transformer architecture could be used to improve the interpretability of resulting clusters, e.g. by integrating an attention-based feature importance score[58]. Second, in our implementation of VaDeSC-EHR, we took some limited steps in representing relations between medical concepts by embedding multiple layers of the ICD-10 ontology, using the alphanumeric ICD-10 codes. However, more sophisticated, and potentially more powerful, approaches have been published that could be integrated with VaDeSC-EHR. Examples include modeling the entire ICD-10 ontology by representing ICD-10s as combinations of their ancestors via an attention mechanism[59] as well as generating embeddings based on ICD-10 text descriptions instead of the ICD-10 codes[60]. Third, depending on the data source used, additional data modalities could be included to provide a more comprehensive view of a patient's disease presentation, as well as to investigate additional factors potentially confounding the interpretation of the clustering, such as additional longitudinal EHR data modalities (e.g. medications, lab tests, and surgical procedures) and demographics (e.g. age, sex, education) or molecular data modalities commonly available in biobanks (e.g. genomics, proteomics)[5].

In our study, we used data from the UK Biobank. The UK Biobank study provides extensive molecular and health data from approximately 500,000 volunteers from the UK. These data are moreover linked to EHR data collected from primary and hospital inpatient care interactions. Like the work that we built upon[3,4], we focused on diagnosis trajectories. Our reason for this is that typical EHR data modalities such as prescribed medications and lab tests are not available from UK Biobank. Additionally, while longitudinal data on procedures are available, these data are only available in the highly UK-specific

OPCS3/4 coding systems that are not easily mappable to internationally used systems and their inclusion would hence lead to highly UK-specific models[61,62]. In addition to the limited availability of EHR data modalities, another drawback of using UK Biobank data for EHR modeling is that the number of individuals covered by UK Biobank is small compared to many EHR databases such as IBM MarketScan[63]. However, the rich multimodality of biobanks such as the UK Biobank, with data ranging from lifestyle, medical history, and biometric data to molecular data such as whole genome sequencing data, provides wide-ranging opportunities for the downstream interpretation of analyses such as presented in our study.

In conclusion, we demonstrated how VaDeSC-EHR, integrating recent advances in risk modeling and EHR modeling, is highly effective at disentangling complex relationships between cluster-specific diagnosis trajectories and survival times. Hence, VaDeSC-EHR can be a powerful tool for supporting the development of precision medicine approaches through its ability to discover novel risk-associated patient subgroups.

## Methods

### EHR dataset from UK Biobank

This study was conducted using the UK Biobank resource, which has ethical approval and its own ethics committee (https://www.ukbiobank.ac.uk/ethics/). This research has been conducted using UK Biobank resources under Application Number 57952. For our analyses, we used both the primary and hospital inpatient care diagnosis records made available via the UK Biobank study[64]. We started from 451,265 patients with available hospital inpatient care data. For each patient, a diagnosis sequence was constructed by interleaving the hospital inpatient care data with any available primary care data based on their timestamps. We then mapped all resulting diagnosis codes from Read v2/3 and ICD-9 to ICD-10[64]. For those codes mapping to multiple ICD-10 codes, we included all possible mappings. Keeping only patients with at least five diagnoses, at most 200 diagnoses, and at least one month of diagnosis history, the resulting dataset consisted of 352,891 patients.

The data processing pipeline is visualized in Supplementary Fig. 16, and the resulting dataset is summarized in Table 1, Supplementary Table 1, and Supplementary Fig. 1.

### VaDeSC-EHR architecture

For implementing VaDeSC-EHR, we integrated VaDeSC into a transformer-based encoder/decoder architecture for learning patient representations (Fig. 1). Here, we describe the architecture and generative process in more detail.

**The generative process.** Adopting the approach of VaDeSC[21], first, a cluster assignment $c \in \{2, \ldots, K\}$ is sampled from a categorical distribution: $p(c) = Cat(\pi)$. Then a latent embedding $z$ is sampled from a Gaussian distribution: $p(z|c) = \mathcal{N}(\mu_c, \sigma_c^2)$. The diagnosis sequence $x$ is generated from $p(x|z)$, which for VaDeSC-EHR is modeled by a transformer-based decoder as described below. Finally, the survival time $t$ is generated by $p(t|z, c)$.

### Transformer-based encoder and decoder

**Embedding.** First, for each ICD-10 code, its subcategory (e.g. K50.1), category (e.g. K50), and block (e.g. K50-K52) were extracted using the algorithm shown in Supplementary Note 1.

Then, the ICD-10 code (subcategory, category, and block), age, and type (primary or hospital) were individually embedded, while adding a sinusoidal position embedding to each of the individual embeddings. The position embedding was based on the visit number of the diagnosis within the diagnosis sequence[6]. Hence, diagnoses originating from the same doctor's visit received the same position embedding. Finally, the individual embeddings were summed to arrive

at the final embedding for each diagnosis (Fig. 2):

$$E_{ICD} = E_{ICD\_block} + E_{ICD\_category} + E_{ICD\_subcategory} \quad (1)$$

$$E_{encoder} = E_{ICD} + E_{age} + E_{type} + E_{position} \quad (2)$$

**Encoder.** The embeddings were fed into a classical transformer encoder[6,7] augmented with a SeqPool layer[65] to consolidate the entire sequence of a patient into a single comprehensive representation for that individual. More specifically, given:

$$X_L = f(X_0) \in \mathbb{R}^{b \times n \times d} \quad (3)$$

where $X_L$ is the output of an $L$ layer transformer encoder $f$, and $b$ is the batch size, $n$ is the sequence length, $d$ is the total embedding dimension. $X_L$ was fed into a linear layer $g(X_L) \in \mathbb{R}^{d \times 1}$, and a softmax activation was applied to the output:

$$X'_L = softmax(g(X_L)^T) \in \mathbb{R}^{b \times 1 \times n} \quad (4)$$

This generated an importance weighting for each input token, which was used as follows[65]:

$$z = X'_L X_L = softmax(g(X_L)^T) \times X_L \in \mathbb{R}^{b \times 1 \times d} \quad (5)$$

By flattening, the output $z \in \mathbb{R}^{b \times d}$ was produced, a summarized embedding for the full patient sequence.

**Decoder.** First, the latent representation was transformed using fully connected layers:

$$X = f\left(W^{(i)} \ldots f\left(W^{(1)} Z^{(1)}\right)\right), Z \in \mathbb{R}^{b \times J}, X \in \mathbb{R}^{b \times LH} \quad (6)$$

Here, $i$ is the number of layers, $b$ the batch size, $L$ the sequence length, $J$ the number of latent variables, $H$ the dimensionality of embedding, and $Z$ the latent representation (Fig. 1).

$X$ was then reshaped to $X_{re}$ ($\mathbb{R}^{b \times LH} \rightarrow \mathbb{R}^{b \times L \times H}$) and fed into the transformer decoder[6,7] to generate the reconstructed ICD embedding $\hat{E}_{ICD} = Decoder(X_{re})$. The reconstructed input embedding for each diagnosis for a given patient was then computed as:

$$E_{decoder} = \hat{E}_{ICD} + E_{age} + E_{type} + E_{position} \quad (7)$$

Here, $E_{age}$, $E_{type}$ and $E_{position}$ were copied over from the input to the transformer encoder.

Finally, the original EHR input sequence (up to the level of the ICD-10 subcategory) was reconstructed from $E_{decoder}$ using a softmax function.

## Evidence lower bound (ELBO)
The loss on the architecture as described above was computed using the ELBO as previously defined for VaDeSC by Manduchi et al.[21]. We briefly present the formula and outline its interpretation, but for more details, we refer the reader to Manduchi et al.[21].

The ELBO of the classic VAE[22] looks as follows:

$$L(x) = \mathbb{E}_{q(z|x)} \log p(x|z) - D_{KL}(q(z|x)||p(z)) \quad (8)$$

Here, $x$ is the input patient diagnosis sequence, $z$ is the latent space for the patient. The first term $p(x|z)$ can be interpreted as the reconstruction loss of the autoencoder. In the second term, $q(z|x)$ is the variational approximation to the intractable posterior $p(z|x)$ and can be seen as regularizing $z$ to lie on a multivariate Gaussian manifold[22].

Adding a cluster indicator as in VaDE[23], the ELBO looks as follows:

$$L(x) = \mathbb{E}_{q(z|x)} \log p(x|z) - D_{KL}(q(z,c|x)||p(z,c)) \quad (9)$$

The first term is the same as above. Analogous to the above, in the second term, $q(z,c|x)$ is the variational approximation to the intractable posterior $p(z,c|x)$ and can be seen as regularizing $z$ to lie on a multivariate Gaussian mixture manifold.

Finally, additionally, including a survival time variable $t$ in the model as in VaDeSC[21], the ELBO looks as follows:

$$L(x,t) = \mathbb{E}_{q(z|x)p(c|z,t)} \log p(x|z) + \mathbb{E}_{q(z|x)p(c|z,t)} \log p(t|z,c) \\ - D_{KL}(q(z,c|x,t)||p(z,c)) \quad (10)$$

The reconstruction loss $p(x|z)$ is calculated as sequence length*-mean cross-entropy.

The survival time $p(t|z,c)$ is modeled by a Weibull distribution and adjusts for right-censoring:

$$p(t|z,c) = f(t)^\delta S(t|z,c)^{1-\delta}$$
$$= \left[\frac{k}{\lambda_c^z}\left(\frac{t}{\lambda_c^z}\right)^{k-1} \exp\left(-\left(\frac{t}{\lambda_c^z}\right)^k\right)\right]^\delta \left[\exp\left(-\left(\frac{t}{\lambda_c^z}\right)^k\right)\right]^{1-\delta} \quad (11)$$

Here, the variable $\delta$ represents the censoring indicator, which is assigned 0 when the survival time of the patient is censored, and 1 in all other cases. For each patient, given by the latent space $z$ and cluster assignment $c$, the uncensored survival time is assumed to follow a Weibull distribution: $f(t) = Weibull(\lambda_c^z, k) = \frac{k}{\lambda_c^z}\left(\frac{t}{\lambda_c^z}\right)^{k-1} \exp\left(-\left(\frac{t}{\lambda_c^z}\right)^k\right)$, where $\lambda_c^z = softplus(z^T \beta_c), \beta_c \in \{\beta_1, \beta_2, \ldots, \beta_K\}$. The censored survival time is then described by the survival function $S(t|z,c) = \exp\left(-\left(\frac{t}{\lambda_c^z}\right)^k\right)$.

For more details and the complete derivation of the VaDeSC ELBO, we refer the reader to Manduchi et al.[21].

## Pre-training and fine-tuning strategy
For the real-world data applications (diabetes and CD), we first pre-trained the transformer encoder on the entire UK Biobank EHR dataset (Table 1, column 3). Following the original BERT study, we used a masked diagnosis learning strategy for pre-training. Specifically, for each patient's diagnosis sequence, we set an 80% probability of replacing a code by [MASK], a 10% probability of replacing a code by a random other code, and the remaining 10% probability of keeping the code unchanged. The ICD-10 embeddings were randomly initialized, and the encoder was trained using an Adam optimizer with default beta1 and beta2.

For selecting our final transformer architecture, we followed the Bayesian hyperparameter optimization strategy as described in the BEHRT study[4]. The best-performing architecture consisted of 6 layers, 16 attention heads, a 768-dimensional latent space, and 1280-dimensional intermediate layers (more details in Supplementary Table 2).

After transformer encoder pre-training, we fine-tuned VaDeSC-EHR end-to-end on the T1D/T2D dataset (Table 1, column 4) and the CD dataset (Table 1, column 5). Details around fine-tuning are described in the following section.

## VaDeSC-EHR in various applications
### Synthetic benchmark
**Data generation.** We used a transformer decoder with random weights to simulate diagnosis sequences (Supplementary Fig. 2). More specifically, let $K$ be the number of clusters, $N$ the number of data points, $L$ the capped sequence length, $H$ the dimensionality of embedding, $D$ the size of vocabulary, $J$ the number of latent variables, $k$ the shape parameter of the Weibull distribution and $p_{cens}$ the

probability of censoring. Then, the data-generating process can be summarized as follows:

1. Let $\pi_c = \frac{1}{K}$, for $1 \leq c \leq K$
2. Sample $c_i \sim Cat(\pi)$, for $1 \leq i \leq N$
3. Sample $\mu_{c,j} \sim unif(-10, 10)$, for $1 \leq c \leq K$ and $1 \leq j \leq J$
4. Sample $z_i \sim \mathcal{N}(\mu_{c_i}, \Sigma_{c_i})$, for $1 \leq i \leq N$
5. Sample $seq_i \sim unif(0, L)$, for $1 \leq i \leq N$
6. Let $g_{res}(z) = reshape(ReLU(wz + b), L \times H)$, where $w \in \mathbb{R}^{LH \times J}$ and $b \in \mathbb{R}^{LH}$ random matrices and vectors.
7. Let $x_i = g_{res}(z_i)$, for $1 \leq i \leq N$
8. Let $g_{att}(x) = softmax(\frac{(w_Q x + b_Q)(w_K x + b_K)^T}{\sqrt{H}} + mask)(w_V x + b_V)$, where $w_Q$, $w_K$, $w_V$ and $b_Q$, $b_K$, $b_V$ are random matrices and vectors. Mask is based on $seq_i$
9. Let $x_i = g_{att}(g_{att}(g_{att}(x_i)))$, for $1 \leq i \leq N$
10. Let $g_{dec}(x) = softmax(ReLU(wx + b))$, where $w \in \mathbb{R}^{D \times H}$ and $b \in \mathbb{R}^D$ random matrices and vectors.
11. Let $x_i = argmax(g_{dec}(x_i))[1 : seq_i]$, for $1 \leq i \leq N$
12. Sample $\beta_{c,j} \sim unif(-2.5, 2.5)$, for $1 \leq c \leq K$ and $1 \leq j \leq J$
13. Sample $u_i \sim Weibull(softplus(z_i^T \beta_{c_i}), k)$, for $1 \leq i \leq N$
14. Sample $\delta_i \sim Bernoulli(1 - p_{cens})$, for $1 \leq i \leq N$
15. Let $t_i = u_i$, if $\delta_i = 1$, and sample $t_i \sim unif(0, u_i)$ otherwise, for $1 \leq i \leq N$

In our experiments, we fixed $K = 3$, $N = 30000$, $J = 5$, $D = 1998$ (ICD-10 category-level vocabulary), $k = 1$, $p_{cens} = 0.3$, $L = 100$. For the attention operation, we used 3 attention layers with 10 heads in each layer.

**Model training and hyperparameter optimization.** Hyperparameter optimization was done by a grid search on learning rate ({0.1, 0.05, 0.01, 0.005, 0.001}) and weight decay ({0.1, 0.05, 0.01, 0.005, 0.001}), using an Adam optimizer with default beta1 and beta2. For performance estimation, we split the generated data into three parts, one-third for training, one-third for validating, and one-third for testing the model. We repeated the above 5 times (i.e. for 5 randomly generated datasets) to arrive at a robust average performance estimate.

**Real-world diabetes benchmark**
**Data extraction.** To extract the data, we selected patients by the occurrence of ICD-10 codes E10 (T1D) or E11.3 (T2D) in their diagnosis trajectories and labeled patients by the presence of H36 ("Retinal disorders in diseases classified elsewhere"). Because of sample size limitations, no requirement was placed on the minimum number of occurrences of each of E10 and E11.3. In order to avoid ambiguity in the performance estimation, patients with both E10 and E11 in their disease history were excluded, which resulted in the dataset as summarized in Table 4, column 2. In the training data, all occurrences of E10, E11, and H36 (and their children) were deleted to avoid data leakage. The selected patients substantially fitted the validated phenotype definition[66].

**Model training and hyperparameter optimization.** We fine-tuned VaDeSC-EHR end-to-end on the dataset described above, taking our pre-trained encoder as a starting point. Note that in the fine-tuning stage, the decoder could benefit from the pre-trained encoder, because weights were shared between the two. We used nested cross-validation (NCV)[30] with a 4-fold inner loop for hyperparameter optimization and a 5-fold outer loop for performance estimation. Hyperparameters were optimized using Bayesian optimization on the following hyperparameter grid:

- learning rate of the Adam optimizer: {1e-5, 5e-5, 1e-4, 5e-4, 1e-3},
- weight decay parameters of the Adam optimizer: {1e-6, 1e-5, 1e-4, 1e-3, 1e-2, 0.1},
- dimension of latent variables: {5,10,15,20},
- shape parameter of the Weibull distribution: {1, 2, 3, 4, 5},
- dropout rate: {0.1, 0.2, 0.3, 0.4, 0.5},
- number of reshape layers: {1, 2, 3, 4, 5}.

**Application: progression of Crohn's disease toward intestinal obstruction**
**Data extraction.** To extract the Crohn's disease (CD) patient population, we selected patients by requiring at least one occurrence of the ICD-10 code K50 (Crohn's disease) while excluding patients with a K51 diagnosis (ulcerative colitis) and then labeled patients according to the presence of K56 ("Paralytic ileus and intestinal obstruction without hernia"). This resulted in the dataset summarized in Table 4, column 5. In the training data, all occurrences of K50 and K56 (and their children) were deleted to avoid data leakage. The selected patients substantially fitted the validated phenotype definition[66].

**Model training and hyperparameter optimization.** We fine-tuned VaDeSC-EHR end-to-end on the dataset described above, taking our pre-trained encoder as a starting point. Note that in the fine-tuning stage, the decoder could benefit from the pre-trained encoder, because weights were shared between the two. We used the pre-trained encoder as the initial weight. And applied 5-fold cross-validation for hyperparameter optimization. Hyperparameters were optimized using Bayesian optimization, with a hyperparameter search space defined as for the diabetes model, except that we now also needed to optimize the number of clusters {2, …, K} jointly with the other hyperparameters, because a ground-truth clustering was not available ($K = 4$, in this use case). The best combination of hyperparameters (including the number of clusters) was determined by encouraging a low Bayesian information criterion (BIC) and a high concordance index (CI) through maximizing: $\sqrt{CI^2 + (1 - BIC_{norm})^2}$, where the BIC was normalized to the interval [0, 1] (Supplementary Fig. 17, Supplementary Table 3).

**Methods comparison and metrics**
We compared VaDeSC-EHR to a range of baseline methods: variational deep survival clustering with a multilayer perceptron (VaDeSC-MLP), semi-supervised clustering (SSC)[17], survival cluster analysis (SCA)[18], deep survival machines (DSM)[19], and recurrent neural network-based DSM (RDSM)[20], as well as k-means and regularized Cox PH as naïve baselines. In addition, to assess the influence of the survival loss on the eventual clustering, we included VaDeSC-EHR_nosurv, in which the survival loss of VaDeSC-EHR was turned off. Finally, to assess the influence of absolute age on distinguishing between ground-truth clusters, we included VaDeSC-EHR_relage (with age at first diagnosis subtracted from all elements in the age sequence). We used ICD-10-based TF-IDF features as the input for all methods but RDSM and VaDeSC-EHR, which allow for directly modeling sequences of events.

We evaluated the clustering performance of models, when possible, in terms of balanced accuracy (ACC), normalized mutual information (NMI), adjusted Rand index (ARI), and area under the receiver-operating characteristic (AUC). Clustering accuracy was computed by using the Hungarian algorithm for mapping between cluster predictions and ground-truth labels[67]. The statistical significance of performance difference was determined using the Mann–Whitney U test.

For the time-to-event predictions, we used the concordance index (CI) to evaluate the ability of the methods to rank patients by their event risk. Given observed survival times $t_i$, predicted risk scores $\delta_i$, and censoring indicators $\delta_i$, the concordance index was defined as

$$CI = \frac{\sum_{i=1}^{N} \sum_{j=1}^{N} \mathbf{1}_{t_j < t_i} \mathbf{1}_{\eta_j > \eta_i} \delta_j}{\sum_{i=1}^{N} \sum_{j=1}^{N} \mathbf{1}_{t_j < t_i} \delta_j} \tag{12}$$

**Visualization and enrichment analysis of clusters**
The clusters were visualized using UMAP (uniform manifold approximation and projection)[68] with the Jensen–Shannon divergence[69] as a

distance measure: $JSD(P(c|x_i), P(c|x_j))$, where $P(c|x)$ is the distribution across clusters $c$ given a patient $x$. Cluster cohesion was measured using the silhouette coefficient[70].

We assessed the association of the clustering with sex, age, education level, location of UKB recruitment, 4 genetic principal components, fraction of hospital care data, and overall diagnosis sequence length using a Chi-squared test or a one-way ANOVA. We calculated the differential enrichment of diagnoses between clusters in two ways: (1) for individual diagnoses, and (2) for sequences of diagnoses. For the individual diagnoses, we used the ICD-10 codes as provided by the UK Biobank. For the diagnosis sequences, we first mapped the ICD-10 codes to Phecode[71] and CALIBER codes[72], which provided a higher level of abstraction in defining diseases. For each patient, we then identified all, potentially gapped, subsequences of three diagnoses from the EHR data, with the following constraints: (1) for duplicate diagnoses in the diagnosis sequence, we only considered the first one, (2) the subsequence contained a diagnosis of the disease under study (CD) but did not contain the selected risk event (intestinal obstruction).

We assessed the statistical significance of the differential enrichment using logistic regression models predicting patient clusters from subsequence occurrence while adjusting for the effects of age, sex, PC1, recruitment location, and fraction of hospital care data by including these variables as covariates into the model. We corrected the resulting $p$-values for multiple testing using the Benjamini–Hochberg procedure and set a threshold at 0.05 for statistical significance.

### Analysis of smoking behavior
We analyzed the association of smoking behavior with progression toward intestinal obstruction using multivariate Cox regression models, individually testing the hazard ratio of several variables related to smoking behavior:

1. data field 20160 ("Ever smoked"),
2. diagnosis ICD-10 code F17.2 ("Mental and behavioral disorders due to use of tobacco dependence syndrome"), which is commonly interpreted as nicotine dependence[73],
3. data field 20161 ("Pack years of smoking"),
4. data field 20162 ("Pack years adult smoking as a proportion of life span exposed to smoking"),
5. Current smoking status, which we defined as the union of patients identified as current smokers from data fields 1239 ("Current tobacco smoking") and 20116 ("Smoking status")
6. Previous smoking status, which we defined as patients who had ever smoked but are not currently smoking. Additionally, to make sure the patients were not smoking at the time of their first CD diagnosis, we excluded patients whose assessment date was after the date of their first CD diagnosis.

The UK Biobank data fields we used contained data collected between 2006 and 2010.

We adjusted the above models for the effects of age, sex, PC1, recruitment location, fraction of hospital care data, and time difference between the CD onset and the nearest smoking diagnosis (F17.2) or assessment date, by including these variables as covariates in the model.

### Genetic analysis of patient clusters
We computed pathway-based polygenic risk scores ('pathway PRSs' henceforth) using PRSet[74] to assess genetic differences between the patient clusters, restricting ourselves to UK Biobank participants of European ancestry[74]. Quality control steps were performed before calculating pathway PRSs, including filtering of SNPs with genotype missingness > 0.05, minor allele frequency (MAF) < 0.01, and with Hardy–Weinberg Equilibrium (HWE) $p$-value $< 5 \times 10^{-8}$. We focused on

164 biological pathways related to Crohn's disease as retrieved from the Gene Ontology – Biological Process (GO-BP) database, selected based on a literature and keyword search ("IMMUNE") (Table S1)[75,76]. We calculated pathway PRSs for each (patient, pathway) pair using variants located in exon regions. For each pathway PRS, we then fitted a logistic regression model predicting cluster from pathway PRS, while adjusting for age, sex, PC1, recruitment location, and the fraction of hospital care data. The $p$-value of the resulting coefficient was calculated using a log-likelihood ratio test[77] and corrected for multiple testing using the Benjamini–Hochberg procedure. For determining statistical significance, we used a threshold of 0.05.

After the pathway-level analysis, we extracted the individual genetic variants contributing to the pathway PRS. We applied Chi-squared tests to identify the significant SNPs[78] and corrected the resulting $p$-values for multiple testing using the Benjamini–Hochberg procedure. Finally, we fitted logistic regression models predicting cluster from mutation status, while adjusting for confounding by including age, sex, PC1, recruitment location, and the fraction of hospital care data as covariates in the models. We defined mutation status by dominant coding, thereby comparing no copy of the risk allele to at least one copy.

### Reporting summary
Further information on research design is available in the Nature Portfolio Reporting Summary linked to this article.

## Data availability
The data that support the findings of this study are available from UK Biobank (www.ukbiobank.ac.uk). Researchers can apply to use the UK Biobank resource for health-related research that is in the public interest (https://www.ukbiobank.ac.uk/register-apply/). Source data are provided with this paper.

## Code availability
The code that supports the findings of this study is available from GitHub https://github.com/JiajunQiu/VaDeSC-EHR (https://doi.org/10.5281/zenodo.14299831).

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

## Acknowledgements

This research has been conducted using UK Biobank, a major biomedical database (www.ukbiobank.ac.uk). This research has been conducted using UK Biobank resources under Application Number 57952. We thank all the employees from Boehringer Ingelheim – Global Computational Biology and Digital Sciences for their support during the project.

## Author contributions

Model design: J.Q., J.d.J.; method development: J.Q.; data analysis: J.Q., Y.H.; definition of CD application: J.Q., L.L., C.W., J.S., J.d.J.; writing the manuscript: J.Q., J.d.J., A.M.E., I.B.; definition and supervision of research project: J.d.J.; providing assistance to the project: J.A., B.A.B., S.G., P.K., S.M., B.N.; funding the project: Z.D., J.N.J.; All authors read, edited, and approved the article.

## Competing interests

The authors declare no competing interests.
