## [Peer review file · Nature Communications]

Deep representation learning for clustering longitudinal survival data from electronic health records

Corresponding Author: Dr Jiajun Qiu

Version 0:

Reviewer comments:

Reviewer #1

(Remarks to the Author)

This paper explores the use of the VaDeSC model to cluster patients from the UK Biobank suffering from Crohn's disease. In particular, the authors propose integrating transformers in the encoder-decoder architecture of VaDeSC to model EHR sequences, thereby taking into account the patients' disease history and their interaction with event risk.

Q) What are the noteworthy results?

The authors explore the clustering results of Crohn's disease patients in their progression towards intestinal obstruction. I find the results rather insightful, especially the difference in the genetic background between different clusters.

Q) Will the work be of significance to the field and related fields? How does it compare to the established literature? If the work is not original, please provide relevant references.

The proposed work applies the VaDeSC model (<https://openreview.net/forum?id=RQ428ZptQfU>) to a data modality different from the ones used in the original paper. However, the VaDeSC model is a probabilistic approach that does not depend on the specific architecture used for the encoder/decoder. In the original paper, for example, VaDeSC used both MLPs for tabular data and CNNs for images. Therefore, I do not agree with the authors' statement that they "introduce a novel deep probabilistic model for clustering longitudinal time-to-event data," and I would not rename VaDeSC to TransVarSur as that could create confusion for the readers.

On the other hand, I believe that clustering survival patients using EHR sequences is of great relevance for the community and could lead to interesting results.

Q) Does the work support the conclusions and claims, or is additional evidence needed?

The novelty claims are not well-supported, as explained above. Additionally, the UK biobank data results should be further validated by measuring (1) the stability of the clustering results across different runs of the model, and (2) by reporting the results on a held-out test set.

Q) Are there any flaws in the data analysis, interpretation, and conclusions? Do these prohibit publication or require revision?

I believe that for a sound comparison, the authors should only report the results on the test set. However, no information regarding the train-validation-test split is provided in the paper.

Q) Is the methodology sound? Does the work meet the expected standards in your field? Is there enough detail provided in the methods for the work to be reproduced?

The method is sound as it is taken from a previously published paper and the additional architectural changes are well explained in the appendix. I think the authors should instead expand on the data processing steps taken, as this part was not clearly explained.

ADDITIONAL COMMENTS:

I would suggest the authors to re-write the paper by not claiming methodological novelty but rather putting the emphasis on the EHR application.

(Remarks on code availability)

Reviewer #2

(Remarks to the Author)

General Comments:

In this paper, the authors propose a transformer-based model (TransVarSur) for clustering patients using their sequence of time- and location-stamped ICD-10 codes and predicting their risk of specific outcomes of interest. They first validate their model architecture using synthetic data, then show it outperforms baseline models on differentiating Type I vs. Type II Diabetes patients (biologically related but well-separated diseases) and predicting their risk of retinal disorders. They then apply the model to identify four novel subtypes of Crohn's Disease with different rates of progression towards intestinal obstruction. Disease subtyping and risk prediction has enormous potential impact in the field of precision medicine by allowing for individualized treatments that provide the highest patient benefit.

Overall, the manuscript is innovative and significant, though there are some concerns to address prior to publication. The novel combination of EHR data, longitudinal survival information, and a transformer-based architecture provides significantly better patient subtyping and risk prediction than similar state-of-the-art models. The integration of multiple data types (e.g., EHR data and genetic information via principal components analysis) is innovative and helps provide biological relevance alongside EHR data. The hierarchical ICD code analysis may help generalize the model and reduce over-reliance on specific coding practices at certain hospitals. Areas for improvement are detailed below, but generally include further detail on the input data and training process of TransVarSur and how TransVarSur performs on the Crohn's disease dataset when it does not have full survival information available.

Specific Comments:

Major Comments:

1. The authors note in the methods section that they use the classical transformer architecture (6 layers, 16 attention heads, 768-dimensional latent space, etc.). Were other hyperparameter combinations considered or evaluated for optimal performance? Similar transformer models using individual-level time sequences for prediction tasks [Savcic et al., Nat. Comp. Sci. 2024] show that model performance can vary somewhat substantially based on transformer architecture (supplementary table 8 of aforementioned article).
2. While it appears that TransVarSur does indeed extract ground-truth clusters more accurately than existing models (balanced accuracy of 0.64 and 0.81 on the synthetic and T1D/T2D baselines), the clusters do not appear to be well-separated in the UMAP plots for T1D/T2D and for CD. Specifically, in the CD plots, clusters 1, 2, and 3 do not appear to congregate within a similar latent space representation. Moreover, Cluster 2 is shown in Fig. 4B to have the lowest risk-free survival but its patients are spread throughout the UMAP embedding space in Fig. 4A. Perhaps other data embedding methods like PaCMAP [Wang et al., JMLR, 2021] might yield better cluster separation?
3. Further detail on how the number of clusters hyperparameter was chosen to be 4 for the Crohn's Disease analysis could be helpful. This appears straightforward in the synthetic baseline (set to be 3 in the synthetic data) and for T1D/T2D (2 clusters), but additional detail could help elucidate how to identify novel disease clusters when a ground truth is not well established.
4. For the benchmark analysis, it was noted that Type I and Type II diabetes were differentiated by presence of specific ICD-10 codes. Was a single occurrence of these codes considered diagnostic for the specific diabetes phenotype, or were additional constraints on the number of code occurrences or time separation between them required to be included as a case?
5. As noted by the authors, the development of TransVarSur helps promote precision medicine by identifying patient subtypes that may be at higher risk for a specific outcome or disease trajectory. Moreover, the authors demonstrate the performance of TransVarSur_nosurv, which does not use survival information to determine cluster membership or risk trajectories. In precision medicine applications, this may present a more realistic use-case, where predicting a patient's course at time of diagnosis might be the goal. Would it be possible to show how the length of available survival data affects the predictive accuracy and confidence of TransVarSur (e.g. intestinal obstruction risk for a patient diagnosed with Crohn's disease 6 months ago vs 10 years ago)? If not, it may also be beneficial to see how TransVarSur_nosurv performs on the Crohn's Disease dataset and if the identified clusters mirror those identified by TransVarSur.

Minor Comments:

1. Figure 3 is captioned as "Performance on simulated benchmark data". However, it is referenced in the text as relating to the T1D/T2D benchmark and its values align with Table 3 (T1D/T2D benchmark) as opposed to Table 2 (Synthetic benchmark).
2. In the analysis of smoking behavior, it is noted that smoking history is determined via data field 20160 ("ever smoked") and ICD10 code F17 (nicotine dependence). Are any more granular measures available, such as pack-year history of smoking or differentiating between current and former smokers? The last distinction may be especially notable if a patient has a remote history (e.g., quit 30 years ago) of smoking and would flag positive on data field 20160 but who may otherwise be treated as

a current non-smoker.

3. As a quick clarification, is the number of diagnoses in the EHR shown in Figure S4 referring to unique diagnostic codes, or to total diagnosis codes including repeats?
4. A few minor grammatical corrections throughout, including:
 - a. A comma should be added after "a subtype of inflammatory bowel disease (IBD)".
 - b. The Oxford comma is inconsistently applied throughout the article.
 - c. The word "the" can be removed after "In addition to evaluating the cluster predictions."
 - d. The word "to" should be added after "trained for 200 epochs compared."
5. The acronym "TF-IDF" is used to discuss the architecture of VaDeSC but is not spelled out in the main body of the text. Similarly, the acronym "PRS" is spelled out in the methods section (p. 20) but is used in the earlier results section (p. 13).

(Remarks on code availability)

Code and simulation data were downloaded from the GitHub repository provided by the authors. The GitHub directions were followed, though the simulation code could not be run, prompting an error: "where is your optimized hyperparameters?" To note, the operating system of the computer used to evaluate the code was different than the architecture the authors used to develop the code, so the included .yaml file could not be loaded. As a result, packages had to be loaded individually, which may have caused compatibility issues.

Reviewer #3

(Remarks to the Author)

(Remarks on code availability)

Reviewer #4

(Remarks to the Author)

(Remarks on code availability)

Reviewer #5

(Remarks to the Author)

The manuscript Deep representation learning for clustering longitudinal survival data from electronic health records by Qiu et al. aims to stratify patients with complex conditions via joint modeling of disease trajectories and risk events. Building upon the approach described in Manduchi et al. (2022), authors replace the variational autoencoder architecture of the VaDeSC model with a transformer-based architecture. Leveraging their TransVarSur model, authors present results on both synthetic and real-world data, i.e., (1) a synthetic dataset obtained by leveraging the randomly initialized TransVarSur decoder to obtain an EHR sequence of ICD codes and sampling from a Weibull distribution to obtain risk events; (2) EHR ICD code based data for patients with T1D and T2D from the UK Biobank; and (3) EHR ICD code based data for patients with Chron's disease from the UK Biobank.

The scope of this work is incredibly important for precision medicine because disorders, often categorized by the same ICD code, present a heterogeneity of phenotypes and disease courses. Clustering of longitudinal survival data can help disentangle complex disorders by enabling the discovery of clusters of patients based on their clinical histories, while capturing the interactions of such clusters with survival outcomes.

Although the methodology presented is well-supported by previous literature, the experiments reported do not seem to match the study's claims. As such, we believe that the manuscript would benefit from substantial major revisions to clarify its aims and a more transparent presentation of the results, especially regarding the model's performance and the clinical relevance of the findings.

Hereafter both minor and major comments section by section:

Introduction

AI/ML Terminology Use. The term "AI" is quite broad and could be refined to more specific terminology related to the study, such as "language models" and "variational autoencoders" (VAEs), which are types of deep learning (DL) models, a subset

of machine learning (ML). Considering the aim of the study, the authors should focus on introducing the state-of-the-art in DL methods for patients stratification, particularly in the context of language models (LMs) and VAEs, which are relevant to the task at hand.

Architecture. The encoder-decoder architecture is a common framework used in various machine learning models, including VAEs and transformers. Despite their structural similarities, the functions and mechanisms of the encoder-decoder architecture in these models differ significantly. Specifically, while VAEs are primarily focused on reconstructing the input sequence, transformers are designed for sequence-to-sequence tasks, such as text generation. In the Methods section, the TransVarSur architecture is described as incorporating a BERT-like model within an encoder-decoder framework, utilizing a reshaping function that allows it to operate during the decoding phase as well. To enhance the clarity of the manuscript, the authors should provide a more detailed explanation of the implemented architecture and clearly delineate how it differs from the traditional autoencoder-like architecture.

Methods

Disease Trajectories. The term "disease trajectories" may be misleading as the longitudinal health sequences discussed in the study contain only ICD codes, which are more indicative of comorbidity profiles. Electronic Health Records (EHRs) include various other types of information, such as lab tests, medications, and procedures. It is unclear why these additional data types were not included in the analysis. The authors should address this omission and provide a rationale for their data selection criteria.

Innovation. The section The TransVarSur architecture largely reiterates details from Manduchi et al. (2022) [1], making repetition unnecessary except for the SeqPool layer. The authors should instead focus on providing a clearer explanation of how the BERT-like architecture in their model integrates with the generative components to effectively model temporal data, including relevant formulas. Additionally, a detailed description of the reshaping function that allows the architecture to be used during the decoding phase should be provided for clarity.

Architecture. The authors present Figure 1 and Figure 2 in the results section, showcasing the architecture and embedding structure of TransVarSurv, but fail to describe them in detail neither in the result nor in the method section. It is unclear how the EHR sequence is reconstructed via the 'decoder' considering the multi-level hierarchy of the ICD10 code embeddings. More details should be provided.

Information Leakage. When creating the input sequences of real-world data how was information leakage addressed? Not enough details are present (besides the removal of the target diagnoses from the sequences) to prevent the bias of the performance due to ICD terms related to Chron and T2/1D diagnoses and corresponding outcomes. For example, T1D patients were identified with ICD-10 code E.10 and T2D patients were identified via ICD-10 code E.11.3, i.e. lower in code hierarchy compared to T2D code. However, other ICD10 codes, such as E11.65 (Type II diabetes mellitus with hyperglycemia) are not explicitly excluded. It is therefore not clear if and how no information leakage was ensured.

Dataset description. The datasets are incompletely described. No information is provided regarding Chron's Disease dataset. Authors should provide information on how the cohort was identified within UK Biobank dataset, and how the outcome event was defined.

Tasks. In general, it seems that the authors perform a large number of experiments (simulation, detection, subtyping and exploration), but it might be beneficial to concentrate on a smaller number of more detailed experiments. For example, not enough information to understand PRS scores role in the clusters of patients with Chron's disease. Moreover, type I and II diabetes are very different conditions characterized by different epidemiology, clinical course and pathophysiological mechanisms. It is unclear how the ability to distinguish type I and II diabetes patients from EHR translates to the much more complex task of identifying novel disease subtypes. Authors should provide a rationale for choosing this task.

Results

Authors should present more fine-grained statistics on the datasets extracted from the UK Biobank, e.g. what were the inclusion/exclusion criteria for the cohort selection of patients with diabetes and Chron's disease? What's the ratio of patients for whom the risk event was not censored? How many years are the diagnoses trajectories spanning and what is the lag between visits? Such information on patient cohorts should be added to Table 4, including counts of patients with T1D and T2D and sociodemographic factors included in the models as confounders.

Authors should provide more details (e.g., loss function on validation set) for the pretraining and fine-tuning phases of the model. Moreover, a better justification of the hyperparameters used should be given. If not, the authors should consider adopting a hyperparameter tuning strategy to better justify their decisions.

Results for the real-world datasets lack the investigation of the relationship between censoring and clusters.

In the results on the synthetic dataset, authors state that the worse performance on time-to-event prediction is due to the model overfitting to the synthetic dataset. It would be important to have a demonstration of that (loss on training and validation) and experiments to avoid that happening via regularization techniques, or less fine-tuning steps.

The enrichment analysis results should be supported by the literature and not presented as novel findings.

Measures of clusters cohesion and separation (e.g., Silhouette) would better demonstrate whether the statement that the representations of patients are consistent with the clusters in the latent space is sound. From Figure 4A and S1A it seems that the embeddings are not sampled from a cluster-conditioned distribution, which make the reader question on the successful fine-tuning of the model.

[1] Manduchi, Laura, Ričards Marcinkevičs, Michela C. Massi, Thomas Weikert, Alexander Sauter, Verena Gotta, Timothy Müller et al. "A deep variational approach to clustering survival data." arXiv preprint arXiv:2106.05763 (2021).

(Remarks on code availability)

Reviewer #6

(Remarks to the Author)

I co-reviewed this manuscript with one of the reviewers who provided the listed reports. This is part of the Nature Communications initiative to facilitate training in peer review and to provide appropriate recognition for Early Career Researchers who co-review manuscripts

(Remarks on code availability)

Version 1:

Reviewer comments:

Reviewer #1

(Remarks to the Author)

I want to thank the authors for addressing the concerns raised in my initial review. After carefully reviewing the revisions, I am satisfied with the changes in the manuscript, with the exception of the writing style (see additional comments).

Below, I provide an overview of how my concerns have been addressed:

Novelty and Clarity of Methodology: In the original submission, I raised concerns about the claims of methodological novelty, particularly around the use of VaDeSC. The authors have now clarified that their work is an application of VaDeSC in a new domain, rather than a novel methodological contribution. They have also emphasized the integration of transformer-based architectures in EHR modeling, as inspired by BEHRT and Med-BERT, which I find to be a valuable contribution.

Stability of Clustering Results and Test Set Validation: I requested additional validation of the model's performance. The authors have now included results from multiple model runs. Additionally, they implemented a nested cross-validation procedure to ensure the robustness and generalizability of their model, which is now well-documented in the revised manuscript. This addresses my concerns and strengthens the conclusions drawn from the analysis.

Data Processing Details: The original version of the manuscript lacked sufficient details on the data processing steps. I appreciate that the authors have now expanded this section, providing a clearer explanation of how data were handled, including the steps taken to preprocess EHR data and how these data were integrated into the transformer encoder-decoder architecture. This additional detail ensures reproducibility and enhances the overall transparency of the study.

Rewriting to Emphasize EHR Applications: In my previous review, I suggested that the manuscript should focus less on methodological novelty and more on the application to EHR data, as this is where the study's true contribution lies. The authors have successfully rewritten the Introduction and Discussion sections to better align with this focus, which I believe improves the manuscript.

Additional Comments:

I believe that the paper would benefit from minor rewriting as certain sections (especially the ones that have been recently added) are not yet well-written and clear.

In particular, in the new abstract, it is not fully clear what VaDeSCEHR is (lines 21-22), and a sentence to clarify that would help the readability.

line 7, I believe something is missing as the sentence appears not complete: 'In all cases, any resulting clusters were purely the risk event and would suffer from the limitations we outlined above.'

'VaDeSC integrates unsupervised generative clustering in the manner of VaDE': again this sentence should be rewritten for clarity

Table 2: I would use only VaDeSC-MLP and rather clarify that it does not use a transformer in the text

I would like to suggest that the authors adopt the name "VaDeSC-EHR" rather than "VaDeSCEHR" for better readability.

(Remarks on code availability)

Reviewer #2

(Remarks to the Author)

Revision Comments:

I appreciate the authors' thoughtful revisions and responses to our feedback, including both in-text clarifications and

additional experiments as needed. Specifically, the clarification of methods for transformer hyperparameter optimization, updated UMAP projection with Jensen-Shannon distance to better separate clusters, and revision of the model name to clarify its position relative to existing literature addressed points that several reviewers had referenced. Overall, this revision is stronger than the original manuscript. Nevertheless, some points still remain to be addressed by the authors, outlined below.

Specific Comments:

Major Comments:

- 1) The authors provide an explanation for why single ICD code occurrences are considered diagnostic, showing how only 73.5% of cases have two diabetes mellitus diagnoses and only 41.4% have at least three. While this approach of prioritizing sensitivity over specificity for phenotype definitions is reasonable, the phenotype definition could be strengthened by stricter exclusion criteria. The authors note that individuals with both E10 and E11 codes are excluded due to ambiguity, but other exclusion criteria may also be considered, such as E09 (drug induced diabetes), E84 (cystic fibrosis), consistent with specific type 1 diabetes definitions in PheKB (<https://phekb.org/phenotype/type-1-diabetes>). Moreover, use of a validated and citeable phenotype definition (whether from PheKB or other sources) for both diabetes and Crohn's disease could strengthen the results.
- 2) For the experiment distinguishing Type I and Type II diabetes, the authors note that age at first diabetes diagnosis was blinded due to diabetes codes E10 and E11 being removed (p. 8, ll. 23-24). Was any additional blinding to age performed to help prevent information leakage? For example, ICD code R73 (hyperglycemia) may be noted prior to diagnosis of diabetes and could reveal the age of diabetes diagnosis. Would VaDeSCEHR be able to distinguish Type I and Type II diabetes cases if it was blinded to patient age information, for instance by counting time from the first visit?
- 3) One reviewer noted that the pathophysiology, epidemiology, and natural history of Type I and Type II diabetes are substantially different and may not fully represent the task of disease sub-phenotyping. While I agree with the authors' arguments that the baseline models did not perform well in distinguishing between Type I and Type II diabetes (indicating it is a difficult task for traditional models), I am not sure this is sufficient to show that VaDeSCEHR is good at extracting closely-related disease subtypes. Perhaps a subtyping task with more similar (yet distinct) diseases could be considered? A natural example, given the paper's focus on Crohn's disease (CD) subtyping, could be the task of distinguishing CD from ulcerative colitis (UC) within a cohort of all patients with inflammatory bowel disease. Moreover, such an analysis could elucidate whether any UC and CD subtype clusters overlap. Alternatively, an explanation of why such an analysis would not be feasible in this dataset (e.g., unclear labels) could also suffice.
- 4) The CD clusters were shown to have a highly significant association ($p = 3.91e-35$) with UK Biobank recruitment location (p. 12, l. 24). However, this could also simply be due to it is also possible that this is simply due to the relatively high number of categories and the computation of the chi-squared statistic. To provide more intuition into the site-specific differences in cluster distribution, could a table or figure show the proportions of each CD cluster at the different sites, thereby giving some insight into the effect size and qualitative differences between sites? If sites do appear to qualitatively differ in cluster makeup in addition to the quantitative differences already shown, a brief discussion about why this effect may be observed could be helpful. For instance, is the effect due to local differences in ICD coding or, perhaps, to regional differences or environmental causes?
- 5) Some further clarification of the hyperparameter optimization process for cluster number selection in the CD subtyping experiment could help: specifically, what cluster numbers (i.e., K) were considered? Supplementary figure 17 depicts 50 hyperparameter combinations, but the values of the values of the tested hyperparameter combinations are not readily apparent. Would it be possible to present this information in a table like supplementary table 3? Moreover, the authors note that the best combination of hyperparameters was chosen by maximizing the L2 norm of CI and $(1 - BIC)$. However, the red point shown in the graph to be the final hyperparameter combination appears to have a CI of ~ 1 and BIC of ~ 0.85 , which would yield an objective value of ~ 1.011 . In contrast, there are several points near the top and left-hand side of the graph, corresponding to a CI of ~ 0.95 and BIC of ~ 0.05 , yielding an objective value of ~ 1.344 . Is it possible that the x-axis of supplementary figure 17 is mislabeled and should be $1 - BIC$? If so, the red point does seem to maximize the stated criterion.

Minor Comments:

- 1) In the results section (p. 13, ll. 4-7), the authors hypothesize that one cluster may have a lower abundance of intestinal bacteria related to hypertension. Is this hypothesis supported by data available within the UK Biobank (either directly via microbiome analysis or indirectly via related diagnoses), or is this mechanism speculative? If the latter, the authors may consider adding a disclaimer that this is one putative mechanism but the available data cannot support or refute it.
- 2) Baseline models (k-means+Cox PH, SSC, SCA, DSM, RDMS) are introduced via abbreviation in the results (p. 7, l. 13) but are not spelled out until the methods section (p. 21)
- 3) Figure 3 still refers to VaDeSCEHR as TransVarSur in the legend, as well as in subfigure c

Typographical / Grammatical Comments:

- 1) p. 7, l. 17: RDMS is spelled 'RDMS'
- 2) p. 13, l. 35: I believe 'pathways' should be changed to 'pathway'

(Remarks on code availability)

(Remarks to the Author)

(Remarks on code availability)

I was able to download and execute the code from the GitHub repository linked by the authors. Following the instructions in the README file, I was able to successfully train the example model over 200 epochs. However, the code generated an error "[Errno 2] No such file or directory: 'squeue'" after model training. I attempted to create a subdirectory named "squeue" in the top-level directory, but this did not alleviate the aforementioned error. As a caveat, I attempted to run this code on a different operating system than the authors used, which may cause compatibility issues.

Reviewer #4

(Remarks to the Author)

(Remarks on code availability)

Reviewer #5

(Remarks to the Author)

We thank the authors for carefully revising the manuscript which has greatly improved. All our comments have been addressed.

(Remarks on code availability)

Reviewer #6

(Remarks to the Author)

(Remarks on code availability)

Version 2:

Reviewer comments:

Reviewer #2

(Remarks to the Author)

The reviewer appreciates the authors' thoughtful responses and specific addressing of concerns, including the rationale behind using Type 1 and 2 diabetes (as opposed to other diseases), the inclusion of VaDeSC-EHR_noage, updating of the statistical test to assess differences in phenotype distribution by recruitment site, and tabulation of the hyperparameter testing process. Overall, this manuscript has been substantially improved. The authors identified suitable computable phenotypes for diabetes and Crohn's disease for their dataset and noted they used these phenotypes in their response; however, one remaining minor concern is that the methods section for data extraction (p. 20, ll. 2-9, 24-29) should be updated to reflect the use of these computable phenotypes. Barring this slight re-writing, my previous concerns have been adequately addressed and no new concerns have arisen.

(Remarks on code availability)

REVIEWER COMMENTS

Reviewer #1 (Remarks to the Author):

This paper explores the use of the VaDeSC model to cluster patients from the UK Biobank suffering from Crohn's disease. In particular, the authors propose integrating transformers in the encoder-decoder architecture of VaDeSC to model EHR sequences, thereby taking into account the patients' disease history and their interaction with event risk.

We would like to thank the reviewer for taking the time to read our manuscript and provide helpful comments and suggestions. Please allow us to address the comments one by one. Changes to the manuscript are highlighted in yellow.

Q) What are the noteworthy results?

The authors explore the clustering results of Crohn's disease patients in their progression towards intestinal obstruction. I find the results rather insightful, especially the difference in the genetic background between different clusters.

We thank the reviewer for their appreciation of the genetic characterization of the clusters.

Q) Will the work be of significance to the field and related fields? How does it compare to the established literature? If the work is not original, please provide relevant references.

The proposed work applies the VaDeSC model (<https://openreview.net/forum?id=RQ428ZptQfU>) to a data modality different from the ones used in the original paper. However, the VaDeSC model is a probabilistic approach that does not depend on the specific architecture used for the encoder/decoder. In the original paper, for example, VaDeSC used both MLPs for tabular data and CNNs for images. Therefore, I do not agree with the authors' statement that they "introduce a novel deep probabilistic model for clustering longitudinal time-to-event data," and I would not rename VaDeSC to TransVarSur as that could create confusion for the readers.

On the other hand, I believe that clustering survival patients using EHR sequences is of great relevance for the community and could lead to interesting results.

We appreciate the reviewer's observation. We naturally agree with the reviewer that VaDeSC is independent from the specific encoder/decoder architecture, and hence our approach is an application of VaDeSC. We have edited the text to more accurately reflect this and have rewritten the text to put more emphasis on EHR modeling applications.

While VaDeSC is indeed an important component of our model, we should note that similarly we built upon a growing body of literature around using transformers for EHR modeling (mainly BEHRT [1] and Med-BERT [2]) to develop a custom encoder/decoder architecture. We feel this is a non-trivial part of our work that was perhaps underemphasized in the original version of the manuscript, and hence we have edited the text (mainly the Methods section: page 16-17) to better reflect this. Given the

substantial importance of both VaDeSC and transformer-based EHR modeling, we have renamed TransVarSur to VaDeSCEHR, and renamed the other occurrences of VaDeSC (with a multi-layer perceptron encoder/decoder architecture) to VaDeSC-MLP. We hope this more accurately reflects the dependencies, the architecture-independent nature of VaDeSC and the more application-oriented focus of our work.

[1] Li Y, Rao S, Solares JRA, Hassaine A, Ramakrishnan R, Canoy D, Zhu Y, Rahimi K, Salimi-Khorshidi G. BEHRT: Transformer for Electronic Health Records. *Sci Rep.* 2020 Apr 28;10(1):7155. doi: 10.1038/s41598-020-62922-y

[2] Rasmy L, Xiang Y, Xie Z, Tao C, Zhi D. Med-BERT: pretrained contextualized embeddings on large-scale structured electronic health records for disease prediction. *NPJ Digit Med.* 2021 May 20;4(1):86. doi: 10.1038/s41746-021-00455-y

Q) Does the work support the conclusions and claims, or is additional evidence needed?

The novelty claims are not well-supported, as explained above. Additionally, the UK biobank data results should be further validated by measuring (1) the stability of the clustering results across different runs of the model, and (2) by reporting the results on a held-out test set.

We thank the reviewer for the useful suggestions and note that, as per the reviewer's comment, we addressed novelty claim above. To address the remaining two points:

(1) We repeated the Crohn's disease and T1D/T2D benchmark with 10 differently randomly initialized embeddings and weights. We observed stable clustering among the 10 runs and have included the results in Supplementary Fig. 4 and Supplementary Fig. 7 (page 8, line 24-26; page 12, line 16-17).

(2) We now included a nested cross-validation approach to assess the generalizability of the model, with a 4-fold inner CV loop for hyperparameter optimization and a 5-fold outer CV loop for performance estimation, reporting the average of the 5 resulting performance figures as the final performance estimate in Table 3 and Fig. 3 (Method: page 20, line 13-25; Results: page 8, line 1-38; page 9, line 1-8).

Q) Are there any flaws in the data analysis, interpretation, and conclusions? Do these prohibit publication or require revision?

I believe that for a sound comparison, the authors should only report the results on the test set. However, no information regarding the train-validation-test split is provided in the paper.

As discussed above, we have now included a nested-cross validation procedure for estimating generalizability.

Q) Is the methodology sound? Does the work meet the expected standards in your field? Is there enough detail provided in the methods for the work to be reproduced?

The method is sound as it is taken from a previously published paper and the additional architectural changes are well explained in the appendix. I think the authors should instead expand on the data processing steps taken, as this part was not clearly explained.

We thank the reviewer for the useful suggestion and have now included more details around data processing in the Methods section (Supplementary Fig. 16, page 16, line 2-12), as well as more details around how the transformer encoder and decoder process the input data (page 16-17).

ADDITIONAL COMMENTS:

I would suggest the authors to re-write the paper by not claiming methodological novelty but rather putting the emphasis on the EHR application.

As outlined above, we have now rewritten the manuscript, especially in the Introduction and Discussion sections to emphasize the application based on VaDeSC and transformer-based EHR modeling (BEHRT/Med-BERT-like architectures) (page 3, line 25-28 and page 14, line 36-39).

Reviewer #2 (Remarks to the Author):

General Comments:

In this paper, the authors propose a transformer-based model (TransVarSur) for clustering patients using their sequence of time- and location-stamped ICD-10 codes and predicting their risk of specific outcomes of interest. They first validate their model architecture using synthetic data, then show it outperforms baseline models on differentiating Type I vs. Type II Diabetes patients (biologically related but well-separated diseases) and predicting their risk of retinal disorders. They then apply the model to identify four novel subtypes of Crohn's Disease with different rates of progression towards intestinal obstruction. Disease subtyping and risk prediction has enormous potential impact in the field of precision medicine by allowing for individualized treatments that provide the highest patient benefit.

Overall, the manuscript is innovative and significant, though there are some concerns to address prior to publication. The novel combination of EHR data, longitudinal survival information, and a transformer-based architecture provides significantly better patient subtyping and risk prediction than similar state-of-the-art models. The integration of multiple data types (e.g., EHR data and genetic information via principal components analysis) is innovative and helps provide biological relevance alongside EHR data. The hierarchical ICD code analysis may help generalize the model and reduce over-reliance on specific coding practices at certain hospitals. Areas for improvement are detailed below, but generally include further detail on the input data and training process of TransVarSur and how TransVarSur performs on the Crohn's disease dataset when it does not have full survival information available.

We would like to thank the reviewer for taking the time to read our manuscript and provide helpful comments and suggestions. Please allow us to address the comments one by one. Changes to the manuscript are highlighted in yellow.

(Please note that in the revised manuscript, TransVarSur has been renamed to VaDeSCEHR)

Specific Comments:

Major Comments:

1. The authors note in the methods section that they use the classical transformer architecture (6 layers, 16 attention heads, 768-dimensional latent space, etc.). Were other hyperparameter combinations considered or evaluated for optimal performance? Similar transformer models using individual-level time sequences for prediction tasks [Savcicens et al., Nat. Comp. Sci. 2024] show that model performance can vary somewhat substantially based on transformer architecture (supplementary table 8 of aforementioned article).

We thank the reviewer for their comment. We have indeed considered and evaluated other hyperparameter combinations. More specifically, we applied a Bayesian hyperparameter optimization strategy as used in the BEHRT study [1] to select the model used in our work. To more clearly reflect the work done here, we have edited the Methods section and included a table of architectures and their respective generalization performance in the supplementary information (Supplementary Table 3) (page 19, line 1-4).

[1] Li Y, Rao S, Solares JRA, Hassaine A, Ramakrishnan R, Canoy D, Zhu Y, Rahimi K, Salimi-Khorshidi G. BEHRT: Transformer for Electronic Health Records. Sci Rep. 2020 Apr 28;10(1):7155. doi: 10.1038/s41598-020-62922-y.

2. While it appears that TransVarSur does indeed extract ground-truth clusters more accurately than existing models (balanced accuracy of 0.64 and 0.81 on the synthetic and T1D/T2D baselines), the clusters do not appear to be well-separated in the UMAP plots for T1D/T2D and for CD. Specifically, in the CD plots, clusters 1, 2, and 3 do not appear to congregate within a similar latent space representation. Moreover, Cluster 2 is shown in Fig. 4B to have the lowest risk-free survival but its patients are spread throughout the UMAP embedding space in Fig. 4A. Perhaps other data embedding methods like PaCMAP [Wang et al., JMLR, 2021] might yield better cluster separation?

We thank the reviewer for their useful suggestion. We agree that the visualization was not optimal. Hence, we have explored a number of additional approaches, including the suggested PaCMAP approach. In the end, we achieved the best results using UMAP with a customized distance calculation (Jensen–Shannon distance; for a demo please refer to https://github.com/JiajunQiu/VaDeSCEHR/blob/main/UMAP_JS.ipynb). We added the relevant details to the Methods section (Methods: Visualization and enrichment analysis of clusters) (page 21, line 22-25) and updated Fig. 4a and Supplementary Fig. 5a.

3. Further detail on how the number of clusters hyperparameter was chosen to be 4 for the Crohn's Disease analysis could be helpful. This appears straightforward in the synthetic baseline (set to be 3 in

the synthetic data) and for T1D/T2D (2 clusters), but additional detail could help elucidate how to identify novel disease clusters when a ground truth is not well established.

The number of clusters $\{1, 2, \dots, K\}$ was optimized jointly with the other hyperparameters. Different combinations of hyperparameters (number of clusters, the dimension of latent variables, the shape parameter of the Weibull distribution, `weight_decay`, learning rate, etc) were evaluated in a Bayesian hyperparameter optimization setting. The best combination of hyperparameters (including the number of clusters) was determined by encouraging a low Bayesian information criterion (BIC) and a high concordance index (CI) through maximizing: $\sqrt{CI^2 + (1 - BIC_{norm})^2}$, where the BIC was normalized to the interval $[0, 1]$. (page 20, line 33-44).

For example, for the CD application we evaluated 50 hyperparameter combinations using Bayesian hyperparameter optimization and identified a hyperparameter combination with $K=4$ as the best one. We have further clarified this in the Methods sections of the manuscript and included an additional figure on this as Supplementary Fig. 17.

4. For the benchmark analysis, it was noted that Type I and Type II diabetes were differentiated by presence of specific ICD-10 codes. Was a single occurrence of these codes considered diagnostic for the specific diabetes phenotype, or were additional constraints on the number of code occurrences or time separation between them required to be included as a case?

In the manuscript, a single occurrence of these codes was indeed considered diagnostic, with a side note that we excluded patients who had both E10 (T1D) and E11 (T2D) in their disease history, in order to arrive at a dataset with unambiguous class labels that could be used for technically validating VaDeSCEHR on real-world data, instead of simulated data.

We understand that imposing additional requirements can be beneficial and reduce the noise in our dataset. However, there are two main reasons for requiring only a single occurrence:

- 1) The UK Biobank has a small sample size ($\sim 0.5E6$) compared to typical EHR databases. Imposing more stringent requirements on the number of occurrences or time separation between them would make the current study impossible. For example, increasing the occurrence by just one, without considering time separation, would already result in a loss of over 25% of samples: 73.5% of patients have at least two diabetes mellitus diagnoses (either E10 or E11) and 41.1% have at least three.
- 2) The purpose was to design a proof-of-principle experiment with unambiguous ground truth labels that would allow us to demonstrate how our approach compares to other methods on actual real-world data, instead of only on simulated data. The clinical relevance here was less important.

We have clarified the above in the Methods section of the manuscript (page 20, line 5-12).

5. As noted by the authors, the development of TransVarSur helps promote precision medicine by identifying patient subtypes that may be at higher risk for a specific outcome or disease trajectory. Moreover, the authors demonstrate the performance of TransVarSur_nosurv, which does not use

survival information to determine cluster membership or risk trajectories. In precision medicine applications, this may present a more realistic use-case, where predicting a patient's course at time of diagnosis might be the goal. Would it be possible to show how the length of available survival data affects the predictive accuracy and confidence of TransVarSur (e.g. intestinal obstruction risk for a patient diagnosed with Crohn's disease 6 months ago vs 10 years ago)? If not, it may also be beneficial to see how TransVarSur_nosurv performs on the Crohn's Disease dataset and if the identified clusters mirror those identified by TransVarSur.

We thank the reviewer for their interesting suggestions. We naturally agree that predicting a patient's disease course only from data available at time of first diagnosis ("baseline" data) is a very realistic use case for the application of precision medicine approaches in clinical practice. In our work, as noted by the reviewer, we instead focus on the joint retrospective analysis and clustering of diagnosis trajectories and event risk profiles. However, in this setting, methodologies such as VaDeSCEHR can provide an important first step in the development of precision medicine approaches by supporting the identification of novel subgroup-specific and targetable disease mechanisms. Following the identification of a potentially targetable patient subgroup, predictive modeling could be used for prognostic enrichment of clinical trials [1], which can eventually influence treatment decisions in clinical practice, through a drug label reflecting the enrichment strategies used to select patients in the clinical trials [2]. We have added a few sentences to the Discussion section to outline these applications (page 14, line 29-35; page 15, line 4-7). Additionally, we conducted some experiments to answer the reviewer's remaining questions.

To address the first question (how the length of available survival data affects the predictive accuracy and confidence), we conducted two regression analyses:

- 1) Predicting the absolute risk prediction error from the time difference between first CD diagnosis date and intestinal obstruction date.
- 2) Predicting the absolute prediction error from the length of the available diagnosis data.

The absolute risk prediction error increased only slightly (but significantly) with increasing no. of years between intestinal obstruction and CD onset. The absolute risk prediction error did not increase significantly with increasing length of available diagnosis data.

To address the second question (how TransVarSur_nosurv performs on the Crohn’s Disease dataset), we clustered CD patients using VaDeSCEHR_nosurv (the new name of TransVarSur_nosurv). Here, we would however like to stress that the purpose of implementing VaDeSCEHR was to jointly cluster diagnosis trajectories and event risk profiles. Switching off the survival loss term (VaDeSCEHR_nosurv) will per definition lead to a different clustering of patients, because the model is fit to a different objective function. More specifically, the model will have no specific incentive anymore to identify clusters that are very different in terms of their event risk profiles.

Indeed, from the results above, we can see that VaDeSCEHR_nosurv (TransVarSur_nosurv) clusters the CD patients into two clusters (first two panels), compared to the original four clusters reported in the manuscript. Directly comparing the VaDeSCEHR and VaDeSCEHR_nosurv clusterings in a cross table (heatmap plot), we see that VaDeSCEHR splits VaDeSCEHR_nosurv cluster 1 roughly into three clusters: cluster1 (26%), cluster3 (35%) and cluster 4 (37%), which are characterized by very different risk profiles (see manuscript). Meanwhile, VaDeSCEHR roughly splits VaDeSCEHR_nosurv cluster 2 mainly into two clusters: cluster3 (37%) and cluster 4 (52%), which are similarly characterized by different risk profiles. However, VaDeSCEHR_nosurv’s clusters, while significantly different overall, do not display qualitatively very different risk profiles compared to VaDeSCEHR’s clusters.

[1] Birkenbihl, C., de Jong, J., Yalchik, I. & Fröhlich, H. Deep learning-based patient stratification for prognostic enrichment of clinical dementia trials. medRxiv, 2023.2011.2025.23299015 (2023). <https://doi.org:10.1101/2023.11.25.23299015>

[2] U.S. Food and Drug Administration. Enrichment Strategies for Clinical Trials to Support Approval of Human Drugs and Biological Products - Guidance for Industry. (2019). <https://doi.org:https://www.fda.gov/regulatory-information/search-fda-guidance-documents/enrichment-strategies-clinical-trials-support-approval-human-drugs-and-biological-products>

Minor Comments:

1. Figure 3 is captioned as “Performance on simulated benchmark data”. However, it is referenced in the text as relating to the T1D/T2D benchmark and its values align with Table 3 (T1D/T2D benchmark) as opposed to Table 2 (Synthetic benchmark).

We have revised the manuscript based on the comments above.

2. In the analysis of smoking behavior, it is noted that smoking history is determined via data field 20160 (“ever smoked”) and ICD10 code F17 (nicotine dependence). Are any more granular measures available, such as pack-year history of smoking or differentiating between current and former smokers? The last distinction may be especially notable if a patient has a remote history (e.g., quit 30 years ago) of smoking and would flag positive on data field 20160 but who may otherwise be treated as a current non-smoker.

We thank the reviewer for their valuable comments and suggestions. We added a number of analyses to the manuscript to address them.

First, we added a analyses with “Pack years of smoking” (UK Biobank data field: 20161) and “Pack years adult smoking as proportion of life span exposed to smoking” (UK Biobank data field: 20162) as Supplementary Fig. 15a,b. These two variables were similarly associated with slower progression towards intestinal obstruction within cluster 1. (Method: page 22, line 9-10, Results: page 13, line 22-24).

Second, we have replaced our analysis of F17 with an analysis of F17.2. F17.2 can be considered to more accurately reflect nicotine dependence [1]. To account for potentially distant histories of smoking, we now include the time difference between CD onset and the nearest F17.2 diagnosis as a covariate in our model (Method: page 22, line 7-8 and line 20-22; Results: page 13, line 18-22).

Third, we added an analysis on current smokers by defining a patient as a current smoker if they reported current smoking at least one of the two data-fields 1239 (Current tobacco smoking) and 20116 (Smoking status). Additionally, we adjusted for potential confounding by including the time difference between CD onset and the assessment date as a covariate in our models (Supplementary Fig. 15c,d).

Fourth, we added an analysis on previous smokers, by defining previous smokers as patients who had ever smoked but are not currently smoking (within the 2006 – 2010 UK Biobank assessment time window). Additionally, in order to make sure the patients were not smoking at the time of their first CD diagnosis, we excluded patients whose assessment date was after the date of their first CD diagnosis (Supplementary Fig. 15e,f).

All above analyses confirmed the initial observation that smoking was associated with slower progression towards obstruction in cluster 1.

We hope the above addresses the reviewer's concerns.

[1] Huttunen R, Heikkinen T, Syrjänen J. Smoking and the outcome of infection. *J Intern Med.* 2011 Mar;269(3):258-69. doi: 10.1111/j.1365-2796.2010.02332.x

3. As a quick clarification, is the number of diagnoses in the EHR shown in Figure S4 referring to unique diagnostic codes, or to total diagnosis codes including repeats?

Supplementary Fig. 8f (formerly Figure S4) refers to the total number of diagnosis codes including repeats. We have clarified this in the main text on page 12, line 25.

4. A few minor grammatical corrections throughout, including:

- a. A comma should be added after "a subtype of inflammatory bowel disease (IBD)".
- b. The Oxford comma is inconsistently applied throughout the article.
- c. The word "the" can be removed after "In addition to evaluating the cluster predictions."
- d. The word "to" should be added after "trained for 200 epochs compared."

We have revised the manuscript based on the comments above.

5. The acronym "TF-IDF" is used to discuss the architecture of VaDeSC but is not spelled out in the main body of the text. Similarly, the acronym "PRS" is spelled out in the methods section (p. 20) but is used in the earlier results section (p. 13).

We have spelled out both occurrences in the main text (page 6, line 33; page 13, line 35).

Reviewer #2 (Remarks on code availability):

Code and simulation data were downloaded from the GitHub repository provided by the authors. The GitHub directions were followed, though the simulation code could not be run, prompting an error: "where is your optimized hyperparameters?" To note, the operating system of the computer used to evaluate the code was different than the architecture the authors used to develop the code, so the included .yaml file could not be loaded. As a result, packages had to be loaded individually, which may have caused compatibility issues.

We appreciate it that the reviewer took the time of testing the code. We fixed the bug the pushed the code to the repository. And it is now in <https://github.com/JiajunQiu/VaDeSCEHR>.

Reviewer #3 (Remarks to the Author):

Reviewer #4 (Remarks to the Author):

Reviewer #5 (Remarks to the Author):

The manuscript Deep representation learning for clustering longitudinal survival data from electronic health records by Qiu et al. aims to stratify patients with complex conditions via joint modeling of disease trajectories and risk events. Building upon the approach described in Manduchi et al. (2022), authors replace the variational autoencoder architecture of the VaDeSC model with a transformer-based architecture. Leveraging their TransVarSur model, authors present results on both synthetic and real-world data, i.e., (1) a synthetic dataset obtained by leveraging the randomly initialized TransVarSur decoder to obtain an EHR sequence of ICD codes and sampling from a Weibull distribution to obtain risk events; (2) EHR ICD code based data for patients with T1D and T2D from the UK Biobank; and (3) EHR ICD code based data for patients with Chron's disease from the UK Biobank.

The scope of this work is incredibly important for precision medicine because disorders, often categorized by the same ICD code, present a heterogeneity of phenotypes and disease courses. Clustering of longitudinal survival data can help disentangle complex disorders by enabling the discovery of clusters of patients based on their clinical histories, while capturing the interactions of such clusters with survival outcomes.

Although the methodology presented is well-supported by previous literature, the experiments reported do not seem to match the study's claims. As such, we believe that the manuscript would benefit from substantial major revisions to clarify its aims and a more transparent presentation of the results, especially regarding the model's performance and the clinical relevance of the findings.

We would like to thank the reviewer for taking the time to read our manuscript and provide helpful comments and suggestions. Please allow us to address the comments one by one. Changes to the manuscript are highlighted in yellow.

(Please note that in the revised manuscript, TransVarSur has been renamed to VaDeSCEHR)

Hereafter both minor and major comments section by section:

Introduction

AI/ML Terminology Use. The term "AI" is quite broad and could be refined to more specific terminology related to the study, such as "language models" and "variational autoencoders" (VAEs), which are types of deep learning (DL) models, a subset of machine learning (ML). Considering the aim of the study, the authors should focus on introducing the state-of-the-art in DL methods for patients stratification, particularly in the context of language models (LMs) and VAEs, which are relevant to the task at hand.

We thank the reviewer for the useful suggestions. We revised the Introduction section according to the suggestions. Specifically, we replaced occurrences of "AI" and/or "ML" with "DL", and extended the introduction with a discussion of state-of-the-art DL methods such as LLMs and transformers for healthcare data, with applications such as risk prediction and disease progression modeling (page 2, line 12-27). We now also discuss VAEs in some detail, as well as VaDE, an extension to the original VAE formulation that uses a Gaussian mixture prior for enforcing a latent structure that promotes clustering (page 3, line 5-19).

Architecture. The encoder-decoder architecture is a common framework used in various machine learning models, including VAEs and transformers. Despite their structural similarities, the functions and mechanisms of the encoder-decoder architecture in these models differ significantly. Specifically, while VAEs are primarily focused on reconstructing the input sequence, transformers are designed for sequence-to-sequence tasks, such as text generation. In the Methods section, the TransVarSur architecture is described as incorporating a BERT-like model within an encoder-decoder framework, utilizing a reshaping function that allows it to operate during the decoding phase as well. To enhance the clarity of the manuscript, the authors should provide a more detailed explanation of the implemented architecture and clearly delineate how it differs from the traditional autoencoder-like architecture.

We thank the reviewer for the helpful suggestions. In the Methods section of the revision, we provided a more detailed explanation of the architecture and clarified how it differs from the traditional autoencoder-like architecture. Moreover, we provided a clearer explanation of how VaDeSC is integrated into the transformer-based encoder/decoder architecture (page 16, line 23-37; page 17, line 1-32).

Methods

Disease Trajectories. The term "disease trajectories" may be misleading as the longitudinal health sequences discussed in the study contain only ICD codes, which are more indicative of comorbidity profiles. Electronic Health Records (EHRs) include various other types of information, such as lab tests, medications, and procedures. It is unclear why these additional data types were not included in the analysis. The authors should address this omission and provide a rationale for their data selection criteria.

We thank the reviewer for the useful comments and suggestions. We agree that the term "disease trajectories" is not specific enough. Hence, in the revision we have replaced each occurrence of "disease trajectory" with "diagnosis trajectory".

The reason for excluding lab tests and prescriptions/medications is that such longitudinal EHR data are not available from the UK Biobank study. Additionally, while longitudinal data on procedures are available, these data are only available in the highly UK-specific OPCS3/4 coding systems that are not easily mappable to internationally used systems [1,2] and their inclusion would hence lead to highly UK-specific models. Finally, we should note that a focus on diagnosis data is very common in the literature, even in cases where data on lab tests, prescriptions/medications and procedures are in fact available. Examples include BEHRT [3], Med-BERT [4], sEHR-CE [5] and the work of Davide Placido [4], among others. Although the specific reasons are not stated in the respective manuscripts, they are likely similar to the above. Nevertheless, we recognize the importance of including other EHR data modalities to provide a more complete picture of a patient's health status but will leave this to future work. We have clarified the above in the Discussion section (page 15, line 25-39).

[1] Stroganov O, Fedarovich A, Wong E, Skovpen Y, Pakhomova E, Grishagin I, Fedarovich D, Khasanova T, Merberg D, Szalma S, Bryant J. Mapping of UK Biobank clinical codes: Challenges and possible solutions. *PLoS One*. 2022 Dec 16;17(12):e0275816. doi: 10.1371/journal.pone.0275816

[2] Papez V, Moinat M, Voss EA, Bazakou S, Van Winzum A, Peviani A, Payralbe S, Kallfelz M, Asselbergs FW, Prieto-Alhambra D, Dobson RJB, Denaxas S. Transforming and evaluating the UK Biobank to the OMOP Common Data Model for COVID-19 research and beyond. *J Am Med Inform Assoc*. 2022 Dec 13;30(1):103-111. doi: 10.1093/jamia/ocac203. Erratum in: *J Am Med Inform Assoc*. 2023 Apr 19;30(5):1006. doi: 10.1093/jamia/ocad032

[3] Li Y, Rao S, Solares JRA, Hassaine A, Ramakrishnan R, Canoy D, Zhu Y, Rahimi K, Salimi-Khorshidi G. BEHRT: Transformer for Electronic Health Records. *Sci Rep*. 2020 Apr 28;10(1):7155. doi: 10.1038/s41598-020-62922-y

[4] Munoz-Farre, A., Rose, H. & Cakiroglu, S. A. sEHR-CE: Language modelling of structured EHR data for efficient and generalizable patient cohort expansion. *ArXiv abs/2211.17121* (2022)

[5] Placido D, Yuan B, Hjaltelin JX, Zheng C, Haue AD, Chmura PJ, Yuan C, Kim J, Umeton R, Antell G, Chowdhury A, Franz A, Brais L, Andrews E, Marks DS, Regev A, Ayandeh S, Brophy MT, Do NV, Kraft P, Wolpin BM, Rosenthal MH, Fillmore NR, Brunak S, Sander C. A deep learning algorithm to predict risk of pancreatic cancer from disease trajectories. *Nat Med*. 2023 May;29(5):1113-1122. doi: 10.1038/s41591-023-02332-5

Innovation. The section The TransVarSur architecture largely reiterates details from Manduchi et al. (2022) [1], making repetition unnecessary except for the SeqPool layer. The authors should instead focus on providing a clearer explanation of how the BERT-like architecture in their model integrates with the generative components to effectively model temporal data, including relevant formulas. Additionally, a detailed description of the reshaping function that allows the architecture to be used during the decoding phase should be provided for clarity.

Architecture. The authors present Figure 1 and Figure 2 in the results section, showcasing the architecture and embedding structure of TransVarSurv, but fail to describe them in detail neither in the result nor in the method section. It is unclear how the EHR sequence is reconstructed via the 'decoder'

considering the multi-level hierarchy of the ICD10 code embeddings. More details should be provided.

We thank the reviewer for the helpful suggestions. Please allow us to jointly address these two points here. In the Methods section of the revision, we removed unnecessary details on the VaDeSC loss function, and provided a clearer explanation of how VaDeSC is integrated into the transformer-based encoder/decoder architecture. Finally, we added a detailed description of the reshaping function, as well as of the architecture and embedding structure of VaDeSCEHR that should make clear how the EHR sequence is reconstructed (page 16, line 23-37; page 17, line 1-32).

Information Leakage. When creating the input sequences of real-world data how was information leakage addressed? Not enough details are present (besides the removal of the target diagnoses from the sequences) to prevent the bias of the performance due to ICD terms related to Chron and T2/1D diagnoses and corresponding outcomes. For example, T1D patients were identified with ICD-10 code E.10 and T2D patients were identified via ICD-10 code E.11.3, i.e. lower in code hierarchy compared to T2D code. However, other ICD10 codes, such as E11.65 (Type II diabetes mellitus with hyperglycemia) are not explicitly excluded. It is therefore not clear if and how no information leakage was ensured.

We thank the reviewer for their comment. In processing the diabetes data, we deleted all ICD10 codes in the E10 (T1D), E11 (T2D) and H36 (the risk event / outcome retinal disorders) categories from the input diagnoses sequence to avoid information leakage. For example, we deleted all occurrences of E11.65 because E11.65 falls in the E11 category. Because we deleted H36 and all codes below H36 (the risk event) category, the survival time ($T_{H36} - T_{E10 \text{ or } E11}$) was not available to the model during training.

In the Crohn's disease application, it was not necessary to delete the Crohn's disease diagnosis (K50), because all patients included are Crohn's disease patients. The risk event / outcome K56 (intestinal obstruction) and all codes below K56 were however deleted. Hence, the survival time ($T_{K56} - T_{K50}$) was not available to the model during training.

We have clarified the above in the relevant sections in the revised manuscript (Method: page 20, line 5-12; page 20, line 27-32; Results: page 8, line 23-24).

Dataset description. The datasets are incompletely described. No information is provided regarding Chron's Disease dataset. Authors should provide information on how the cohort was identified within UK Biobank dataset, and how the outcome event was defined.

We thank the reviewer for their suggestion. We have now included a table in the Result section describing the UK Biobank data and the extracted CD and diabetes cohorts (Table 1). Additionally, we have added details in the Methods section on how these cohorts were extracted from UK Biobank and how the risk/outcome events were defined (Supplementary Fig. 6, Method: page 20, line 27-32, Results: page 12, line 13-14).

Tasks. In general, it seems that the authors perform a large number of experiments (simulation, detection, subtyping and exploration), but it might be beneficial to concentrate on a smaller number of more detailed experiments. For example, not enough information to understand PRS scores role in the clusters of patients with Chron's disease. Moreover, type I and II diabetes are very different conditions characterized by different epidemiology, clinical course and pathophysiological mechanisms. It is unclear how the ability to distinguish type I and II diabetes patients from EHR translates to the much more complex task of identifying novel disease subtypes. Authors should provide a rationale for choosing this task.

We thank the reviewer for their valuable suggestions. We agree with the reviewer that we do many experiments, and that it would be beneficial to our work to better clarify the purpose of experiments in some cases, as well as to concentrate on a number of smaller more detailed experiments in other cases.

The first two experiments (on synthetic data and on T1D/T2D data) serve to technically validate the approach by VaDeSCEHR's ability to re-cover ground truth clusters labels. Their clinical relevance is of less importance, but they are necessary to establish confidence in the methodology. Contrary to the technical validation experiments, the last experiment (the CD application) is intended to be clinically relevant, but due to the lack of ground truth cluster labels can only be validated through supporting literature.

We defined the second technical validation experiment (T1D/T2D) using real-world data because after a validation with synthetic data (the first technical validation experiment), questions can remain regarding the performance on real-world data. We specifically chose T1D/T2D because, in addition to some similarities ([1, 2]), clear and known differences in clinical course and pathophysiological mechanisms exist between these two subtypes of diabetes [3]. We expected machine learning algorithms to be able to exploit these differences to various degrees in their attempt to recover the ground truth cluster labels (which were obviously not available during training). Indeed, substantial differences in performance were observed between VaDeSCEHR and the baseline methods. While the task of distinguishing T1D from T2D may seem not too complex, the substantial performance differences between the various methods (with some methods performing no/hardly better than random, e.g. SCA, DSM, RDSM, SSC, kmeans+CoxPH) demonstrate that it is far from trivial to recover the ground truth cluster labels from only the EHR sequences, even for state-of-the-art machine learning methods. We believe this illustrates the suitability of diabetes as a real-world data technical validation experiment for VaDeSCEHR. We have edited the main text to clarify the purpose of both validation experiments, and specifically the reason for choosing diabetes (page 6, line 11-17; page 8, line 10-22).

The CD application is our most extensive experiment. We agree with the reviewer that this analysis could be more focused and detailed. In our revised manuscript, we have moved one entire figure plus one figure panel to the supplements, re-organized the remaining results, and added a more in-depth analysis of the genetic/PRS results. More specifically, we tested the 150 variants contributing to the significant pathway PRS and identified SNP rs2523608 as an interesting variant significantly associated with fast progression towards intestinal obstruction (clusters 2 and especially 3). We have further highlighted some links of this SNP with various gastrointestinal disorders as well as with impaired response to interferon beta therapy, which was previously investigated as a potential treatment for Crohn's disease (CD) (Fig.6b,c, page 14, line 8-22).

We hope the above addresses the reviewer's concerns.

[1] Nyaga DM, Vickers MH, Jefferies C, Fadason T, O'Sullivan JM. Untangling the genetic link between type 1 and type 2 diabetes using functional genomics. *Sci Rep*. 2021 Jul 6;11(1):13871.

[2] Inshaw JRJ, Sidore C, Cucca F, Stefana MI, Crouch DJM, McCarthy MI, Mahajan A, Todd JA. Analysis of overlapping genetic association in type 1 and type 2 diabetes. *Diabetologia*. 2021 Jun;64(6):1342-1347.

[3] Zaccardi F, Webb DR, Yates T, Davies MJ. Pathophysiology of type 1 and type 2 diabetes mellitus: a 90-year perspective. *Postgrad Med J*. 2016 Feb;92(1084):63-9. doi: 10.1136/postgradmedj-2015-133281

Results

Authors should present more fine-grained statistics on the datasets extracted from the UK Biobank, e.g. what were the inclusion/exclusion criteria for the cohort selection of patients with diabetes and Chron's disease? What's the ratio of patients for whom the risk event was not censored? How many years are the diagnoses trajectories spanning and what is the lag between visits? Such information on patient cohorts should be added to Table 4, including counts of patients with T1D and T2D and sociodemographic factors included in the models as confounders.

We thank the reviewer for their useful suggestions. In the revised manuscript, we added more details around the inclusion/exclusion criteria that were used for extracting the cohorts to the Methods section. We moved Table 4 (now Table 1) to the Results section, and added statistics on censoring, patient counts, and sociodemographic factors included in the models as confounders. We added data on diagnosis trajectory length and lag between visits as Supplementary Fig. 1 (median and IQR in Table 1), and data on recruitment location as Supplementary Table 1. Finally, we added the distribution of the first genetic principal component as Supplementary Fig. 9a.

Authors should provide more details (e.g., loss function on validation set) for the pretraining and fine-tuning phases of the model. Moreover, a better justification of the hyperparameters used should be given. If not, the authors should consider adopting a hyperparameter tuning strategy to better justify their decisions.

We have indeed considered and evaluated other hyperparameter combinations. More specifically, we applied a Bayesian hyperparameter optimization strategy as used in the BEHRT study [1] to select the model used in our work. To more clearly reflect the work done here, we have edited the Methods section and included a table of architectures and their respective generalization performance in the supplementary information. We described detailed pretraining strategy and related hyperparameter optimization in page 18, line 28-36 and page 19, line 1-7 (Supplementary Table 3). For the fine-tuning part, we provided detailed information in Method section: VaDeSCEHR in various applications (page 19-20).

[1] Li Y, Rao S, Solares JRA, Hassaine A, Ramakrishnan R, Canoy D, Zhu Y, Rahimi K, Salimi-Khorshidi G. BEHRT: Transformer for Electronic Health Records. Sci Rep. 2020 Apr 28;10(1):7155. doi: 10.1038/s41598-020-62922-y.

Results for the real-world datasets lack the investigation of the relationship between censoring and clusters.

We have now included an analysis of the association between censoring and cluster membership, using multinomial logistic regression, while adjusting for potential confounding by including age, sex, PC1, recruitment location, fraction of hospital care data as covariates into the model. The fastest progressing cluster 2 was found most strongly enriched for intestinal obstruction (Supplementary Fig. 8a).

In the results on the synthetic dataset, authors state that the worse performance on time-to-event prediction is due to the model overfitting to the synthetic dataset. It would be important to have a demonstration of that (loss on training and validation) and experiments to avoid that happening via regularization techniques, or less fine-tuning steps.

We apologize for the confusing way in which we tried to make our point in the original manuscript. The model does not overfit, as we can see from the plot below of the survival loss as a function of the number of epochs (overall loss in Supplementary Fig. 3a). Rather, VaDeSCEHR simply performs slightly worse on risk prediction than VaDeSC-MLP in the current setting. In the revised manuscript, we rephrased the corresponding section as follows: Given the known regularizing effects of variational loss terms, it is likely that VaDeSCEHR's risk prediction performance could be improved (possibly at the expense of some clustering performance) by weighting of the different loss terms (e.g. beta-VAE).

The enrichment analysis results should be supported by the literature and not presented as novel findings.

We have added references supporting the enrichment analysis results, highlighted in yellow.

Measures of clusters cohesion and separation (e.g., Silhouette) would better demonstrate whether the statement that the representations of patients are consistent with the clusters in the latent space is sound. From Figure 4A and S1A it seems that the embeddings are not sampled from a cluster-conditioned distribution, which make the reader question on the successful fine-tuning of the model.

[1] Manduchi, Laura, Ričards Marcinkevičs, Michela C. Massi, Thomas Weikert, Alexander Sauter, Verena Gotta, Timothy Müller et al. "A deep variational approach to clustering survival data." arXiv preprint arXiv:2106.05763 (2021).

We thank the reviewer for their helpful suggestion. We agree that the visualization of the clusters using UMAP did not reflect the cluster separation very well. Keeping the limitations of any dimensionality reduction methodology in mind, we explored a number of additional approaches. In the end, we achieved the best results using UMAP with a customized distance calculation (Jensen–Shannon distance; for a demo please refer to https://github.com/JiajunQiu/VaDeSCEHR/blob/main/UMAP_JS.ipynb). We added the relevant details to the Methods section (Methods: Visualization and enrichment analysis of clusters) (page 21, line 22-25) and updated Fig. 4a and Supplementary Fig. 5a (original Figure S1A). Additionally, following the reviewer's suggestion, we calculated silhouette coefficients for our models (0.783 shown in Fig. 4a and 0.964 shown in Supplementary Fig. 5a). Below, we also show the UMAP and survival plots of the unfine-tuned model for the Crohn's disease application (Fig. 4). The unfine-tuned model corresponds to a silhouette coefficient of 0.427 (compared to 0.783 for the fine-tuned model, Fig. 4a) and a concordance index of 0.6 (compared to 0.91 for the fine-tuned model, Fig. 4b), demonstrating successful clustering as well as fine-tuning.

Reviewer #6 (Remarks to the Author):

I co-reviewed this manuscript with one of the reviewers who provided the listed reports. This is part of the Nature Communications initiative to facilitate training in peer review and to provide appropriate recognition for Early Career Researchers who co-review manuscripts

Reviewer #1 (Remarks to the Author):

I want to thank the authors for addressing the concerns raised in my initial review. After carefully reviewing the revisions, I am satisfied with the changes in the manuscript, with the exception of the writing style (see additional comments).

Below, I provide an overview of how my concerns have been addressed:

Novelty and Clarity of Methodology: In the original submission, I raised concerns about the claims of methodological novelty, particularly around the use of VaDeSC. The authors have now clarified that their work is an application of VaDeSC in a new domain, rather than a novel methodological contribution. They have also emphasized the integration of transformer-based architectures in EHR modeling, as inspired by BEHRT and Med-BERT, which I find to be a valuable contribution.

Stability of Clustering Results and Test Set Validation: I requested additional validation of the model's performance. The authors have now included results from multiple model runs. Additionally, they implemented a nested cross-validation procedure to ensure the robustness and generalizability of their model, which is now well-documented in the revised manuscript. This addresses my concerns and strengthens the conclusions drawn from the analysis.

Data Processing Details: The original version of the manuscript lacked sufficient details on the data processing steps. I appreciate that the authors have now expanded this section, providing a clearer explanation of how data were handled, including the steps taken to preprocess EHR data and how these data were integrated into the transformer encoder-decoder architecture. This additional detail ensures reproducibility and enhances the overall transparency of the study.

Rewriting to Emphasize EHR Applications: In my previous review, I suggested that the manuscript should focus less on methodological novelty and more on the application to EHR data, as this is where the study's true contribution lies. The authors have successfully rewritten the Introduction and Discussion sections to better align with this focus, which I believe improves the manuscript.

We would like to thank the reviewer for helpful comments and suggestions. Please allow us to address the comments one by one. Changes to the manuscript are highlighted in yellow.

Additional Comments:

I believe that the paper would benefit from minor rewriting as certain sections (especially the ones that have been recently added) are not yet well-written and clear.

In particular, in the new abstract, it is not fully clear what VaDeSCEHR is (lines 21-22), and a sentence to clarify that would help the readability.

We have rephrased the sentence in the abstract to provide a very high-level description of the architecture and its purpose.

line 7, I believe something is missing as the sentence appears not complete: 'In all cases, any resulting clusters were purely the risk event and would suffer from the limitations we outlined above.'

'VaDeSC integrates unsupervised generative clustering in the manner of VaDE': again this sentence should be rewritten for clarity

We have rewritten these two sentences.

Table 2: I would use only VaDeSC-MLP and rather clarify that it does not use a transformer in the text

We have edited the table accordingly.

I would like to suggest that the authors adopt the name "VaDeSC-EHR" rather than "VaDeSCEHR" for better readability.

We have replaced all occurrences of "VaDeSCEHR" with "VaDeSC-EHR".

Reviewer #2 (Remarks to the Author):

Revision Comments:

I appreciate the authors' thoughtful revisions and responses to our feedback, including both in-text clarifications and additional experiments as needed. Specifically, the clarification of methods for transformer hyperparameter optimization, updated UMAP projection with Jensen-Shannon distance to better separate clusters, and revision of the model name to clarify its position relative to existing literature addressed points that several reviewers had referenced. Overall, this revision is stronger than the original manuscript. Nevertheless, some points still remain to be addressed by the authors, outlined below.

We would like to thank the reviewer for helpful comments and suggestions. Please allow us to address the comments one by one. Changes to the manuscript are highlighted in yellow.

Specific Comments:

Major Comments:

1) The authors provide an explanation for why single ICD code occurrences are considered diagnostic, showing how only 73.5% of cases have two diabetes mellitus diagnoses and only 41.4% have at least three. While this approach of prioritizing sensitivity over specificity for phenotype definitions is reasonable, the phenotype definition could be strengthened by stricter exclusion criteria. The authors note that individuals with both E10 and E11 codes are excluded due to ambiguity, but other exclusion criteria may also be considered, such as E09 (drug induced diabetes), E84 (cystic fibrosis), consistent with specific type 1 diabetes definitions in PheKB (<https://phekb.org/phenotype/type-1-diabetes>). Moreover, use of a validated and citeable phenotype definition (whether from PheKB or other sources) for both diabetes and Crohn's disease could strengthen the results.

We would like to thank the reviewer for this suggestion. We agree that the use of a validated and citable phenotype definition could strengthen the analysis and results. For the revision, we have considered both PheKB [1] and HDR UK Phenotype Library [2]. For various reasons outlined below, none of the phenotype definitions in PheKB could be matched to our data source. However, we were able to match our data and selected

cohorts to the definitions for type 1 diabetes [3], type 2 diabetes [4] and Crohn's disease [5] reported in the HDR UK Phenotype Library and included these references in the revised manuscript.

As mentioned above, we could not match any of the definitions in PheKB to our data source. Specifically, for type 1 diabetes, there are five exclusion criteria in PheKB [6]:

- Prescription of T2D medications: We are not able to consider medications because no record-level EHR medication data is available for UK Biobank individuals. Only diagnosis and procedure data are linked to UK Biobank.
- Diagnosis of T2D: As described before, we excluded patients with a diagnosis of T2D.
- Diagnosis of cystic fibrosis: In our data, there are no T1D patients with a diagnosis of cystic fibrosis.
- Diagnosis of drug induced diabetes: In our data, there are no T1D patients with a diagnosis of drug induced diabetes.
- Diagnosis of malignant cancer: Malignant cancer was included in the PheKB phenotype definition as an exclusion criterion because of its potential to cause secondary diabetes through damaging pancreatic islets cells [personal communication with the author of the definition, Frank Mentch (mentchf@chop.edu)]. Note that secondary diabetes is not consistently captured across different ICD ontologies and not present in the UKB-linked EHRs. Nonetheless, we decided not to add malignant cancer as an additional exclusion criterion for the following reason: In our dataset, we have 470 patients with a diagnosis of malignant cancer, out of a total of 2324 patients. However, fewer than 1 in 100 cases of new-onset diabetes are caused by cancer [7,8]. Thus, we can reasonably expect only about 5 out of our total of 2324 patients to have diabetes caused by cancer. In order to maximize statistical power for performing a reliable performance estimation and method comparison, we decided to not remove the 470 patients with malignant cancer, but instead accept a minor $\sim 0.2\%$ impurity ($5 / 2324 * 100\%$) in our patients, such that we could have a dataset that is $>25\%$ larger ($470 / (2324 - 470) * 100\%$).

For type 2 diabetes [9] and Crohn's disease [10] many of the remaining inclusion/exclusion criteria in PheKB refer to medications or lab tests. Unfortunately, these data types are not available as part of the UKB-linked EHR data.

Hence, we turned to the HDR UK Phenotype Library, and now cite the reported definitions for type 1 diabetes [3], type 2 diabetes [4] and Crohn's disease [5] in the revised manuscript.

- [1] Kirby JC, Speltz P, Rasmussen LV, Basford M, Gottesman O, Peissig PL, Pacheco JA, Tromp G, Pathak J, Carrell DS, Ellis SB, Lingren T, Thompson WK, Savova G, Haines J, Roden DM, Harris PA, Denny JC. PheKB: a catalog and workflow for creating electronic phenotype algorithms for transportability. *J Am Med Inform Assoc*. 2016 Nov;23(6):1046-1052. doi: 10.1093/jamia/ocv202. Epub 2016 Mar 28. PMID: 27026615; PMCID: PMC5070514.
- [2] Thayer DS, Mumtaz S, Elmessary MA, Scanlon I, Zinnurov A, Coldea AI, Scanlon J, Chapman M, Curcin V, John A, DelPozo-Banos M, Davies H, Karwath A, Gkoutos GV, Fitzpatrick NK, Quint JK, Varma S, Milner C, Oliveira C, Parkinson H, Denaxas S, Hemingway H, Jefferson E. Creating a next-generation phenotype library: the health data research UK Phenotype Library. *JAMIA Open*. 2024 Jun 17;7(2):ooae049. doi: 10.1093/jamiaopen/ooae049. PMID: 38895652; PMCID: PMC11182945.
- [3] Alison K Wright, Evangelos Kontopantelis, Richard Emsley, Iain Buchan, Naveed Sattar, Martin K Rutter, Darren M. Ashcroft. PH717 / 1434 - Type 1 Diabetes. Phenotype Library [Online]. 06 October 2021. Available from: <http://phenotypes.healthdatagateway.org/phenotypes/PH717/version/1434/detail/>. [Accessed 15 October 2024]
- [4] Alison K Wright, Evangelos Kontopantelis, Richard Emsley, Iain Buchan, Naveed Sattar, Martin K Rutter, Darren M. Ashcroft. PH718 / 1436 - Type 2 Diabetes. Phenotype Library [Online]. 06 October 2021. Available from: <http://phenotypes.healthdatagateway.org/phenotypes/PH718/version/1436/detail/>. [Accessed 15 October 2024]
- [5] Kuan V, Denaxas S, Gonzalez-Izquierdo A, Direk K, Bhatti O, Husain S, Sutaria S, Hingorani M, Nitsch D, Parisinos C, Lumbers T, Mathur R, Sofat R, Casas JP, Wong I, Hemingway H, Hingorani A. PH144 / 288 - Crohn's disease. Phenotype Library [Online]. 06 October 2021. Available from: <http://phenotypes.healthdatagateway.org/phenotypes/PH144/version/288/detail/>. [Accessed 15 October 2024]
- [6] Huiqi Qu, Jeff Roizen, Frank Mentch, John Connolly, Heather Hain, Patrick Sleiman, Hakon Hakonarson . CHOP. Type 1 Diabetes. PheKB; 2021 Available from: <https://phekb.org/phenotype/1548>
- [7] <https://www.cancer.gov/news-events/cancer-currents-blog/2021/pancreatic-cancer-diabetes-early-detection>
- [8] Mellenthin C, Balaban VD, Dugic A, Cullati S. Risk Factors for Pancreatic Cancer in Patients with New-Onset Diabetes: A Systematic Review and Meta-Analysis. *Cancers (Basel)*. 2022 Sep 26;14(19):4684. doi: 10.3390/cancers14194684. PMID: 36230607; PMCID: PMC9563634.

[9] Josh Denny and Melissa Basford. Vanderbilt University. Type 2 Diabetes - Demonstration Project. PheKB; 2012 Available from: <https://phekb.org/phenotype/73>

[10] Josh Denny and Melissa Basford. Vanderbilt University. Crohn's Disease - Demonstration Project. PheKB; 2012 Available from: <https://phekb.org/phenotype/77>

2) For the experiment distinguishing Type I and Type II diabetes, the authors note that age at first diabetes diagnosis was blinded due to diabetes codes E10 and E11 being removed (p. 8, ll. 23-24). Was any additional blinding to age performed to help prevent information leakage? For example, ICD code R73 (hyperglycemia) may be noted prior to diagnosis of diabetes and could reveal the age of diabetes diagnosis. Would VaDeSCEHR be able to distinguish Type I and Type II diabetes cases if it was blinded to patient age information, for instance by counting time from the first visit?

We want to thank the reviewer for their suggestion. We now included an analysis (VaDeSC-EHR_noage) that blinds the age information by subtracting the age at first diagnosis from the entire diagnosis sequence such that all diagnosis sequences start at age 0. This analysis demonstrated almost identical performance to VaDeSC-EHR, indicating that VaDeSC-EHR primarily utilizes the time intervals between diagnoses, rather than the age itself. We have added the related content at page 8, line 30-36, page 21, line 8-10, and updated Fig. 3 and Table 3.

3) One reviewer noted that the pathophysiology, epidemiology, and natural history of Type I and Type II diabetes are substantially different and may not fully represent the task of disease sub-phenotyping. While I agree with the authors' arguments that the baseline models did not perform well in distinguishing between Type I and Type II diabetes (indicating it is a difficult task for traditional models), I am not sure this is sufficient to show that VaDeSCEHR is good at extracting closely-related disease subtypes. Perhaps a subtyping task with more similar (yet distinct) diseases could be considered? A natural example, given the paper's focus on Crohn's disease (CD) subtyping, could be the task of distinguishing CD from ulcerative colitis (UC) within a cohort of all patients with inflammatory bowel disease. Moreover, such an analysis could elucidate whether any UC and CD subtype clusters overlap. Alternatively, an explanation of why such an analysis would not be feasible in this dataset (e.g., unclear labels) could also suffice.

We thank the reviewer for this thoughtful suggestion. We would like to address a few key points in response to this comment.

First, although we considered IBD subtyping as a benchmark, we found it infeasible due to reasons related to selecting a proper risk event. In the UK Biobank dataset, although UC patients exhibit a wide variety of complications (in terms of ICD-10 codes), these complications are generally rare, leading to high right censoring rates when used as a risk event. Below is a table illustrating censoring rates for common UC complications (from among all 1545 possible risk events that we checked).

code	description	right censoring rate	missing rate
C18	Malignant neoplasm of colon	0.98	0.97
C19	Malignant neoplasm of rectosigmoid junction	1.00	1.00
C20	Malignant neoplasm of rectum	0.99	0.99
D50	Iron deficiency anemia	0.92	0.89
D52	Folate deficiency anemia	1.00	0.99
D62	Acute posthemorrhagic anemia	1.00	1.00
D64	Other anaemias	0.91	0.88
H20	Iridocyclitis	0.99	0.98
H44	Disorders of globe	0.99	0.99
I26	Pulmonary embolism	0.98	0.97
K52	Other and unspecified noninfective gastroenteritis and colitis	0.75	0.60
K56	Paralytic ileus and intestinal obstruction without hernia	0.94	0.92
K57	Diverticular disease of intestine	0.85	0.79
K60	Fissure and fistula of anal and rectal regions	0.97	0.95
K61	Abscess of anal and rectal regions	0.98	0.98
K62	Other diseases of anus and rectum	0.74	0.56
K63	Other diseases of intestine	0.78	0.73
K65	Peritonitis	0.99	0.99
L51	Erythema multiforme	1.00	1.00
L52	Erythema nodosum	1.00	0.99
L88	Pyoderma gangrenosum	1.00	0.99
M05	Rheumatoid arthritis with rheumatoid factor	1.00	1.00
M07	Enteropathic arthropathies	0.99	0.99
R11	Nausea and vomiting	0.92	0.89

Censoring rates as reported in the table above lead to very imbalanced datasets and limit the statistical power in comparing the different methods, making IBD less suitable for a benchmark. Especially strictly supervised methods potentially suffer even more in such a highly imbalanced setting, because there is no unsupervised loss component that may be able to learn a good clustering despite the lack of survival labels. This could lead to comparisons that could be considered unfair. In contrast, the T1D/T2D dataset with a censoring rate of 42%, provides a more balanced and statistically powerful

comparison between methods. We have added a comment on class balance and statistical power to the manuscript (page 8, line 19-21).

We have considered several other options as well, but did not identify a dataset that we found more suitable than diabetes specifically as a benchmark (although they may well be suitable as an eventual application of the method):

- Lung cancer: The subtypes NSCLC and SCLC are not covered by the ICD-10 ontology.
- Parkinson's disease: The main subtypes Tremor-Dominant, Akinetic-Rigid and Juvenile are not covered by the ICD-10 ontology.
- Asthma: After excluding overlaps, very few patients remain for the subtypes allergic and nonallergic asthma: 527 for allergic and 81 for nonallergic asthma.
- Psychiatric disorders and diseases: "No single psychiatric diagnosis has reliably shown to represent a discrete taxa" [1].

Finally, in making a case for diabetes as a benchmark, we would like to note that indeed, presently, the two diabetes subtypes and the differences in underlying mechanisms are well-understood. This has however not always been the case. Historically, type 1 and type 2 diabetes were grouped together as "diabetes mellitus", largely because of the shared clinical symptoms: increased thirst and frequent urination, extreme fatigue, unexplained weight loss, blurred vision, slow healing of wounds and frequent infections. Only less than a century ago, in 1936, Harold Himsworth proposed that some diabetes patients may in fact have insulin resistance rather than insulin deficiency [2]. Moreover, only around the 1970s and 1980s, the ADA (American Diabetes Association) and WHO began consistently using the two subtypes of diabetes [3,4,5]. With VaDeSC-EHR, we try to identify patient clusters based on clinical (ICD-10) profiles only, but with the eventual aim of identifying novel and divergent underlying disease mechanisms. In our opinion, this makes diabetes an almost perfect benchmark:

- Shared clinical symptoms but well-characterized differential disease mechanisms.
- A large number of patients.
- A clinically relevant, frequent and shared risk event.

We hope that the above addresses the reviewer's concern.

References:

[1] Hengartner MP, Lehmann SN. Why Psychiatric Research Must Abandon Traditional Diagnostic Classification and Adopt a Fully Dimensional Scope: Two Solutions to a Persistent Problem. *Front Psychiatry*. 2017 Jun 7;8:101. doi: 10.3389/fpsy.2017.00101. PMID: 28638352; PMCID: PMC5461269.

[2] Polonsky KS. The past 200 years in diabetes. *N Engl J Med.* 2012 Oct 4;367(14):1332-40. doi: 10.1056/NEJMra1110560. PMID: 2303402

[3] Bliss, M. *The Discovery of Insulin.* (University of Chicago Press, 2013)

[4] Diabetes mellitus. Report of a WHO Study Group. *World Health Organ Tech Rep Ser.* 1985;727:1-113. PMID: 3934850.

[5] Classification and diagnosis of diabetes mellitus and other categories of glucose intolerance. National Diabetes Data Group. *Diabetes.* 1979 Dec;28(12):1039-57. doi: 10.2337/diab.28.12.1039. PMID: 510803.

4) The CD clusters were shown to have a highly significant association ($p = 3.91e-35$) with UK Biobank recruitment location (p. 12, l. 24). However, this could also simply be due to it is also possible that this is simply due to the relatively high number of categories and the computation of the chi-squared statistic. To provide more intuition into the site-specific differences in cluster distribution, could a table or figure show the proportions of each CD cluster at the different sites, thereby giving some insight into the effect size and qualitative differences between sites? If sites do appear to qualitatively differ in cluster makeup in addition to the quantitative differences already shown, a brief discussion about why this effect may be observed could be helpful. For instance, is the effect due to local differences in ICD coding or, perhaps, to regional differences or environmental causes?

We thank the reviewer for this interesting observation. We agree with the reviewer and considered specifically that the p-value could be inflated due to outliers in the cell count distribution. Indeed, in our contingency table, 20% of cell counts are below 5, with some cells even having zero counts, violating some general assumptions/guidelines around applying the Chi-squared test, e.g. [1]. Hence, we replaced the parametric Chi-squared test with a non-parametric permutation test, which is more robust under these conditions [2,3]. The new and more robust results indicate no significant association between cluster and location, and we have updated our statement accordingly.

[1] McHugh ML. The chi-square test of independence. *Biochem Med (Zagreb)*. 2013;23(2):143-9. doi: 10.11613/bm.2013.018. PMID: 23894860; PMCID: PMC3900058.

[2] Camargo A, Azuaje F, Wang H, Zheng H. Permutation - based statistical tests for multiple hypotheses. *Source Code Biol Med*. 2008 Oct 21;3:15. doi: 10.1186/1751-0473-3-15. PMID: 18939983; PMCID: PMC2611984.

[3] Lehmann, E.L., Romano, J.P. (2022). *Permutation and Randomization Tests*. In: *Testing Statistical Hypotheses*. Springer Texts in Statistics. Springer, Cham. https://doi.org/10.1007/978-3-030-70578-7_17

5) Some further clarification of the hyperparameter optimization process for cluster number selection in the CD subtyping experiment could help: specifically, what cluster numbers (i.e., K) were considered? Supplementary figure 17 depicts 50 hyperparameter combinations, but the values of the values of the tested hyperparameter combinations are not readily apparent. Would it be possible to present this information in a table like supplementary table 3? Moreover, the authors note that the best combination of hyperparameters was chosen by maximizing the L2 norm of CI and $(1 - \text{BIC})$. However, the red point shown in the graph to be the final hyperparameter combination appears to have a CI of ~ 1 and BIC of ~ 0.85 , which would yield an objective value of ~ 1.011 . In contrast, there are several points near the top and left-hand side of the graph, corresponding to a CI of ~ 0.95 and BIC of ~ 0.05 , yielding an objective value of ~ 1.344 . Is it possible that the x-axis of supplementary figure 17 is mislabeled and should be $1 - \text{BIC}$? If so, the red point does seem to maximize the stated criterion.

Indeed, supplementary figure 17 was mislabeled and should be labeled as $1 - \text{BIC}$. We have corrected supplementary figure 17 and added supplementary table 3 presenting the various hyperparameter combinations that we tested.

Minor Comments:

1) In the results section (p. 13, ll. 4-7), the authors hypothesize that one cluster may have a lower abundance of intestinal bacteria related to hypertension. Is this hypothesis supported by data available within the UK Biobank (either directly via microbiome analysis or indirectly via related diagnoses), or is this mechanism speculative? If the latter, the authors may consider adding a disclaimer that this is one putative mechanism but the available data cannot support or refute it.

This mechanism is indeed speculative, and we have removed it for further streamlining the manuscript.

2) Baseline models (k-means+Cox PH, SSC, SCA, DSM, RDSM) are introduced via abbreviation in the results (p. 7, l. 13) but are not spelled out until the methods section (p. 21)

We have now spelled these out in the results section (page6, line 12-16).

3) Figure 3 still refers to VaDeSCEHR as TransVarSur in the legend, as well as in subfigure c

We have corrected Figure 3.

Typographical / Grammatical Comments:

1) p. 7, l. 17: RDSM is spelled 'RDMS'

2) p. 13, l. 35: I believe 'pathways' should be changed to 'pathway'

We have corrected the above two errors.

Reviewer #3 (Remarks to the Author):

Reviewer #3 (Remarks on code availability):

I was able to download and execute the code from the GitHub repository linked by the authors. Following the instructions in the README file, I was able to successfully train the example model over 200 epochs. However, the code generated an error "[Errno 2] No such file or directory: 'squeue'" after model training. I attempted to create a subdirectory named "squeue" in the top-level directory, but this did not alleviate the aforementioned error. As a caveat, I attempted to run this code on a different operating system than the authors used, which may cause compatibility issues.

We appreciate the reviewer running the code and apologize this bug. The “squeue” statement in the code is specific to the system we ran our code on and should have been deleted. It has now been removed.

As an additional clarification: During hyperparameter optimization, we submit jobs to different compute nodes on our compute cluster and use the “squeue” command from slurm to check whether there are jobs that still need to finish. For a simulation run, which is performed only on the local node, this check is unnecessary.

Reviewer #4 (Remarks to the Author):

Reviewer #5 (Remarks to the Author):

We thank the authors for carefully revising the manuscript which has greatly improved. All our comments have been addressed.

Reviewer #6 (Remarks to the Author):

REVIEWERS' COMMENTS

Reviewer #2 (Remarks to the Author):

The reviewer appreciates the authors' thoughtful responses and specific addressing of concerns, including the rationale behind using Type 1 and 2 diabetes (as opposed to other diseases), the inclusion of VaDeSC-EHR_noage, updating of the statistical test to assess differences in phenotype distribution by recruitment site, and tabulation of the hyperparameter testing process. Overall, this manuscript has been substantially improved. The authors identified suitable computable phenotypes for diabetes and Crohn's disease for their dataset and noted they used these phenotypes in their response; however, one remaining minor concern is that the methods section for data extraction (p. 20, ll. 2-9, 24-29) should be updated to reflect the use of these computable phenotypes. Barring this slight re-writing, my previous concerns have been adequately addressed and no new concerns have arisen.

Thanks for your comment. We have added sentence in the methods section for data extraction to reflect that the selected patients substantially fitted the validated phenotype definition.